# Synaptic mechanisms modulate the spatiotemporal dynamics of striatal direct pathway neurons and motor output

John J Marshall[1]*, Jian Xu[1], Nai-Hsing Yeh[1], Seongsik Yun[1], Toshihro Nomura[1], John N Armstrong[1], Jones Parker[1,2,3]*, Anis Contractor[1,2,4]*

[1]Department of Neuroscience, Feinberg School of Medicine, Northwestern University, Evanston, United States; [2]Department of Psychiatry and Behavioral Sciences, Feinberg School of Medicine, Northwestern University, Evanston, United States; [3]Department of Pharmacology, Feinberg School of Medicine, Northwestern University, Evanston, United States; [4]Department of Neurobiology, Weinberg School of Arts and Sciences, Northwestern University, Evanston, United States

*For correspondence:
john-marshall@northwestern.edu
(JJM);
jones.parker@northwestern.edu
(JP);
a-contractor@northwestern.edu
(AC)

## eLife Assessment

Marshall et al describe the effects of altering metabotropic glutamate receptor 5 activity on activity of D1 receptor expressing spiny projection neurons in dorsolateral striatum focusing on two states - locomotion and rest. The authors examine effects of dSPN-specific constitutive mGlu5 deletion in several motor tests to arrive at this finding. Effects of inhibiting the degradation of the endocannabinoid 2-arachidonoyl glycerol are also examined. Overall, this is a **valuable** study that provides **solid** new information of relevance to movement disorders and possibly psychosis.

**Abstract** Striatal spiny-projection neurons (SPNs) integrate glutamatergic inputs from the motor cortex and thalamus with neuromodulatory signals to regulate motor output. In vivo $Ca^{2+}$ imaging has demonstrated that spatially overlapping ensembles of direct and indirect pathway SPNs (dSPNs, iSPNs) are co-active during spontaneous movement. This co-activity is statistically greater among nearby neurons, correlates with behavioral state, and changes in an SPN-type-specific manner under pathological conditions. How this co-activity arises and whether it is important for movement are not well understood. Co-activity likely arises from shared excitatory inputs, the strength of which are scaled by well-characterized mechanisms of synaptic plasticity. Here, we show that the Group I metabotropic glutamate receptor 5 (mGluR5), which regulates the strength of corticostriatal synapses, is a key mediator of behaviorally relevant SPN co-activity. Pharmacological modulation of mGluR5 signaling bidirectionally altered movement and co-activity, but not the absolute level of activity in dSPNs. Targeted deletion of mGluR5 in dSPNs recapitulated the effects on spatiotemporal neural dynamics and movement, consistent with a striatum-specific effect of mGluR5 modulation. Targeted deletion of mGluR5 also produced changes in the synaptic properties of dSPNs. Separate from any effects on overall activity, our results show that excitatory synaptic modulation influences motor function by coordinating the spatial co-activation of dSPNs in vivo.

## Introduction

The dorsal striatum is central to movement initiation and to regulating the kinematics of learned movements (*Klaus et al., 2019*; *Wolff et al., 2022*). Synaptic plasticity in this input structure of the basal ganglia has been demonstrated to be important for motor learning (*Dang et al., 2006*). In

particular, the strength of corticostriatal synapses between the motor cortex and spiny-projection neurons (SPNs) in the dorsolateral striatum (DLS) is strengthened by learning a motor sequence (**Rothwell et al., 2015**) or during the acquisition of a motor skill (**Yin et al., 2009**; **Hwang et al., 2022**). These glutamatergic, corticostriatal inputs form synapses onto two types of SPNs that are distinguished by their expression of dopamine receptor subtypes and downstream axonal projections. The D1 dopamine receptor-expressing, or direct pathway SPNs (dSPNs) project monosynaptically to the output nuclei of the basal ganglia, the substantia nigra pars reticulata (SNr) and globus pallidus internal segment (GPi). By contrast, D2 dopamine receptor-expressing SPNs (iSPNs) project indirectly to basal ganglia output nuclei via the globus pallidus external segment (GPe) and the subthalamic nucleus (STN) (**Gerfen and Surmeier, 2011**). In classical models of the basal ganglia circuit, activity in dSPNs facilitates movement, and activity in iSPNs inhibits movement. Therefore, the relative level of activity between the two populations is predicted to correlate with an animal's motor state (e.g. moving or at rest).

This rate-model framework of striatal function garnered early support from anatomical and pharmacological observations as well as findings from studies investigating the neuropathology of striatum-related neurodegenerative disease (**Nelson and Kreitzer, 2014**). However, the model is inconsistent with more recent observations employing in vivo imaging techniques. In these studies, where the activity of genetically defined iSPNs and dSPNs can be monitored concurrently in awake behaving animals, dSPNs and iSPNs are consistently co-active during movement initiation and procession through movement sequences (**Cui et al., 2013**). More recently, it was shown that dSPNs and iSPNs simultaneously activate with spatially clustered dynamics, and that this spatiotemporal organization is important for motor control (**Barbera et al., 2016**; **Parker et al., 2018**; **Klaus et al., 2017**). Spatiotemporal clustering, or heightened co-activity among proximal SPN pairs, is most pronounced during periods of rest, but also transiently increases during transitions from rest to movement, then subsides as SPNs become more active during movement progression (**Barbera et al., 2016**; **Parker et al., 2018**; **Klaus et al., 2017**; **Yun et al., 2023**). Importantly, dopamine signaling appears to play an important role in sculpting these dynamics. The loss of dopamine diminishes the spatiotemporal coordination of iSPN activity, while heightened dopamine release decorrelates activity in dSPNs and heightens iSPN co-activity (**Barbera et al., 2016**; **Parker et al., 2018**; **Klaus et al., 2017**; **Yun et al., 2023**).

Although the mechanisms that constrain the spatiotemporal coordination of dSPN and iSPN activity are not precisely known, SPNs have hyperpolarized resting membrane potentials that prevent action potential generation in the absence of glutamatergic input (**Wilson and Groves, 1981**; **Wilson and Kawaguchi, 1996**). Given that glutamate is required for SPN activation, and given dopamine's role in modifying the strength of glutamatergic synapses in SPNs, it is likely that synaptic plasticity influences the spatiotemporal dynamics of SPNs by regulating their collective sensitivity to convergent excitatory inputs (**Guo et al., 2015**; **Wall et al., 2013**).

dSPNs and iSPNs receive excitatory input from the cortex and thalamus. Cortico- and thalamostriatal synapses differ in their expression of pre- and post-synaptic neurotransmitter receptors and forms of synaptic plasticity (**Wu et al., 2015**). Extensive in vitro work has demonstrated various mechanistically distinct types of long-term potentiation (LTP) and long-term depression (LTD) at cortical inputs to dSPNs and iSPNs (**Li and Pozzo-Miller, 2020**; **Surmeier et al., 2009**). While the range of physiological and neuromodulatory mechanisms of corticostriatal plasticity are complex, one well-studied type of plasticity is endocannabinoid (eCB) dependent LTD, induced in vitro by moderate to high frequency stimulation of corticostriatal inputs paired with postsynaptic depolarization (**Wu et al., 2015**). Importantly, corticostriatal LTD requires activation of postsynaptic Gq-linked, Group I metabotropic glutamate receptors (mGluRs). mGluR activation causes the mobilization of endocannabinoids (eCBs) from the postsynaptic neuron which act as a retrograde signal by activating presynaptic CB1 receptors to reduce the release probability of glutamatergic corticostriatal synapses (**Gerdeman et al., 2002**). mGluR LTD occurs at corticostriatal synapses in both dSPNs and iSPNs (**Bagetta et al., 2011**) but not at thalamostriatal synapses (**Wu et al., 2015**). eCB release and retrograde signaling can also produce transient modulation of presynaptic release to cause the short-term depression of corticostriatal synapses (**Kreitzer and Malenka, 2007**; **Marshall et al., 2018**), similar to what has been observed at synapses in several other brain regions (**Chevaleyre et al., 2006**). Despite extensive in vitro work, little is known about the circumstances under which eCB-dependent synaptic modulation

occurs in vivo and how it may relate to motor behavior. Moreover, it is unknown how the modulation of corticostriatal synaptic strength regulates the level and/or spatiotemporal dynamics of SPN activity with respect to movement.

In the striatum, mGluR5 has been shown to have a prominent role in mGluR-dependent synaptic plasticity. Therefore, we asked whether mGluR5 signaling contributes to voluntary movement and the dynamics of dSPN activity in the DLS. We found that manipulating mGluR5 signaling had bidirectional effects on movement and the spatiotemporal dynamics of dSPN activity, with minimal effects on their overall levels of activity. Specifically, using in vivo $Ca^{2+}$ imaging, we found that pharmacological inhibition of mGluR5 signaling produced a reduction in spontaneous movement and heightened the spatially clustered patterns of co-activity in dSPNs, whereas positive modulation of mGluR5 signaling had the opposite effects. Consistent with these findings, conditional knockout of mGluR5 in dSPNs (D1 cKO mice) had minimal effects on the levels of dSPN or iSPN activity but specifically increased the spatiotemporal coordination of dSPN activity and reduced spontaneous locomotion. Collectively, these findings suggest that mGluR-mediated synaptic plasticity mechanisms bidirectionally constrain the spatiotemporal coordination, but not the levels of dSPN activity, into states that correlate with increased or decreased locomotor activity. Our study lays the groundwork for dissociating the separable roles of the activity rates versus patterning of SPN activity in motor control and has implications for correcting disruptions of this patterning in basal ganglia-associated diseases.

## Results

### dSPNs exhibit spatially clustered activity that varies with movement

There is a strong correlation between spatiotemporal patterns of SPN activity and motor output (*Barbera et al., 2016*; *Parker et al., 2018*; *Klaus et al., 2017*), but it is not known if synaptic properties modulate this coordinated activity in vivo. To assess the activity of dSPNs during spontaneous motor behavior, we implanted gradient refractive index (GRIN) lenses into the dorsolateral striatum of mice expressing Cre-recombinase in dSPNs under the control of the *Drd1a* promoter. A Cre-dependent AAV to express GCaMP6f was introduced either by a prior stereotaxic injection of virus or by coating the lenses with a silk fibroin film containing the AAV (*Figure 1a*; *Jackman et al., 2018*). The GRIN lens was coupled to a head-mounted miniscope for 1 P imaging of somatic $Ca^{2+}$ events (*Aharoni and Hoogland, 2019*). *Figure 1b* shows an example of a field of view from one mouse and the segmented outlines of neurons after processing of the videos (*see* **Methods**) and examples of individual dSPN $Ca^{2+}$ dynamics (*Figure 1c*). Consistent with prior observations (*Yun et al., 2023*), $Ca^{2+}$ event rates in dSPNs increased during periods of movement (*Figure 1d and e*). The probability of $Ca^{2+}$ events in detected cells (defined as fluorescence change of 2.5x the baseline for each cell, *see* **Methods**) increased during movement (*Figure 1d*) and the $Ca^{2+}$ event rate increased with increasing animal velocity (*Figure 1e*). Previous work has demonstrated that dSPNs and iSPNs are spontaneously co-active in spatially intermingled clusters, a feature that has been proposed to be important for action selection (*Barbera et al., 2016*; *Parker et al., 2018*; *Klaus et al., 2017*; *Shin et al., 2020*). To examine this spatiotemporal coordination and its modulation by movement state, we measured dSPN co-activity using a Jaccard index of dSPN pairs as a function of the Euclidean distance between the cell pairs in the DLS. This measure captures co-activity between neuron pairs (as opposed to co-inactivity) more specifically than a linear correlation coefficient (*Figure 1f*; **Methods**; *Parker et al., 2018*). Across all cell pairs in the field of view for each session, a correlation matrix was generated in which, for each frame of the recording, the cell pair was scored a 1 if both cells were active, or 0 otherwise. These scores for each frame were divided by the number of events in both cells to generate a co-activity score for that pair. *Figure 1f* shows schematized raster plots to demonstrate this process. Then, those values were normalized to averages measured from independently shuffled $Ca^{2+}$ traces for each cell in the recording (using 1000 independent shuffles for each cell trace), to control for changes in mean activity level between conditions (*Figure 1f*). If cells were co-active independently of distance, then a plot of co-activity normalized to the shuffled data as a function of distance would be flat (*Figure 1g*, **top**), while clustered co-activity (i.e. higher among proximal cells) would show a downward-sloping curve (*Figure 1g*, **bottom**). Consistent with prior reports (*Barbera et al., 2016*; *Parker et al., 2018*; *Klaus et al., 2017*), this analysis revealed that co-activity among neuron pairs is markedly higher among proximal cells and decreases with increasing Euclidean distance between

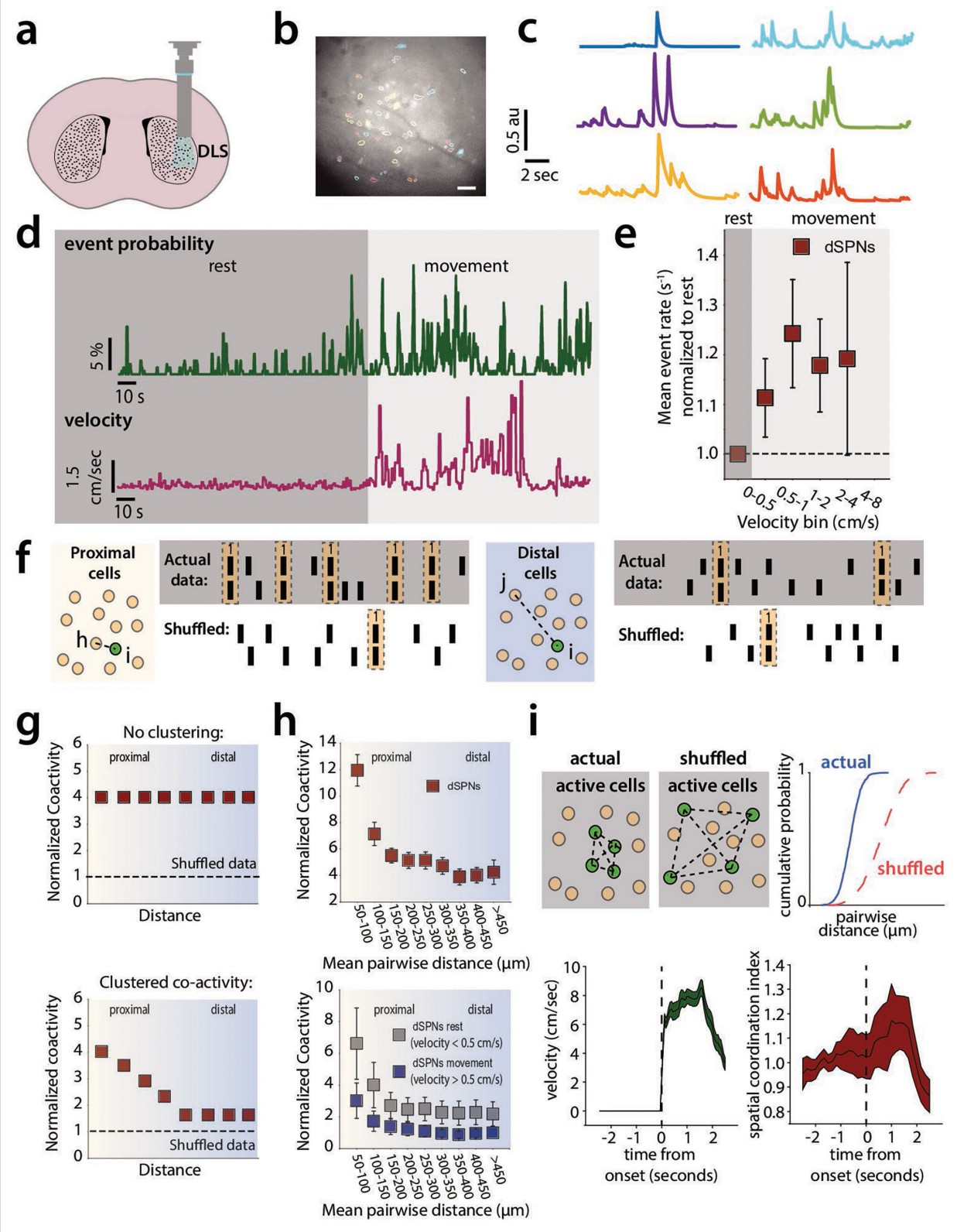

**Figure 1.** Direct spiny-projection neurons (dSPNs) exhibit spatially clustered activity that varies with movement. (**a**) Cartoon representation of implanted gradient refractive index (GRIN) lens and miniscope imaging in the dorsolateral striatum (DLS). (**b**) Examples of segmented cell outlines from the CNMF-E algorithm in a single mouse (scale 100 μm). (**c**) Example of Ca$^{2+}$ traces from corresponding colored and filled segmented cells shown in (**b**). (**d**) The probability of a Ca$^{2+}$ event in detected cells increases during movement in dSPNs. (**e**) The Ca$^{2+}$ event rate in detected cells increases

*Figure 1 continued*

during movement in dSPNs ($p=0.240$, Wilcoxon Signed Rank Test, events/second during bins with velocity >0.5 cm/s vs. during bins with velocity <0.5 cm/s). (**f**) Cartoon schematic to illustrate analysis of co-activity as a function of distance between cell pairs. Across all cell pairs in the field of view for each session, a correlation matrix was generated, in which, for each frame of the recording, the cell pair was scored a 1 if both cells were active, or 0 otherwise. These scores for each frame were averaged across all frames in the recording to generate an average co-activity score for that pair. Then, those values were normalized to averages measured from independently shuffled $Ca^{2+}$ traces for each cell in the recording (using 1000 independent shuffles for each cell trace), to control for changes in mean activity level between conditions. (**g**) Hypothetical plot of shuffle-normalized Jaccard scores versus distance when coactivity is not 'clustered' and hypothetical plot of 'clustered' co-activity. Co-activity decreases during periods of movement (Linear Mixed Effects Model/LMM, $p<0.000$, $z=-4.852$, $\beta=-1.671$, SE=0.344, 95% CI: [-2.346–0.996], N=5 mice). (**i**) The spatial coordination index (SCI) peaks during movement onset ($p<0.000$, $z=12.21$, Wilcoxon Signed Rank Test, comparing the peak value of the normalized SCI during 2.5 s following movement onset to 1). Error bars represent standard error of the mean, $p<0.05$ defines significance.

pairs (*Figure 1h*, **top**). Furthermore, as we had previously demonstrated, co-activity among all pairs was higher during rest and decreased during periods of movement (*Figure 1h*, **top**; *Yun et al., 2023*).

We next quantified the extent to which activity was higher in proximal cells versus distal neurons in each frame of the recording using a spatial coordination index (SCI), as defined previously (*Parker et al., 2018*). Here, for each frame of the down-sampled video, we preserved the cell activity traces but shuffled the pairwise distances for each neuron pair. We then performed a pair of Kolmogorov–Smirnov tests (**methods**) to determine if the cumulative distribution of active cell distances in each frame was smaller (i.e. left-shifted or more 'clustered') or greater (right-shifted or less 'clustered') than that of the shuffled data. This analysis allows for visualization of how the spatiotemporal dynamics of SPN activity change instantaneously (e.g. during movement onset). Consistent with that reported previously, the degree of 'clustered' activity quantified in this way increased significantly during movement onset (within 2.5 s of the mouse initiating movement; *Parker et al., 2018*). This transient increase contrasts with the increased basal co-activity at all cell-cell distances seen during periods when mice were at rest (defined as centroid velocity below a given threshold).

## The spatiotemporal dynamics of dSPN activity is modulated by mGluR5

The dominant form of synaptic plasticity at corticostriatal synapses is mediated by Group 1 mGluRs (*Wu et al., 2015*), and the primary receptor of this class expressed in SPNs is mGluR5 (*Uchigashima et al., 2007*). Germline deletion of mGluR5 causes significant motor phenotypes, but it is not known how mGluR5 signaling directly affects motor function (*Xu et al., 2021*). Due to the known importance of mGluR5 in modulating synaptic weight at corticostriatal synapses, we hypothesized that pharmacological modulation of mGluR5 would affect SPN dynamics in vivo.

To assess the effects of modulating mGluR5 activity on dSPN dynamics during spontaneous behavior, we injected mice with vehicle or the mGluR5 negative allosteric modulator (NAM) fenobam (26 mg/kg via i.p. injection) immediately prior to a 60 min open field test. We analyzed dSPN activity from eight animals expressing GCaMP6f in dSPNs and implanted with GRIN lenses in the DLS (*Figure 2a–c*). We measured activity as mice spontaneously explored the open field arena. On average, we recorded from 91 dSPNs per mouse in these experiments.

Fenobam treatment uniformly reduced locomotion (*Figure 2d and e*), an effect that was driven more by a reduction in movement bout duration than the frequency of movement bouts (*Figure 2f*). Interestingly, dSPN event rates were unchanged following fenobam injection, with $Ca^{2+}$ event rates in dSPNs increasing with spontaneous locomotor speed across the range of running speeds in both fenobam and vehicle-injected animals (*Figure 2g*). Fenobam also had no effect on the amplitude or duration of $Ca^{2+}$ events (*Figure 2—figure supplement 1*).

However, fenobam treatment significantly increased the pairwise co-activity of dSPNs across the whole range of cell-cell distances (*Figure 2h*). Notably, while co-activity at all distances was enhanced by fenobam treatment, we did not resolve any differences in spatial coordination, measured using the SCI, between vehicle and drug treatment as animals transitioned from rest to movement. In both cases, the SCI peaked around the time of movement onset (*Figure 2i*). This suggests that the most significant effect of mGluR5 on dSPN co-activity is to increase its overall extent, rather than the relative extent to which it becomes more or less 'clustered' during behavioral state transitions. As fenobam treatment reduced movement, and co-activity is modulated by movement, we separated our analysis into periods of movement and rest to compare equivalent behavioral states during vehicle and drug

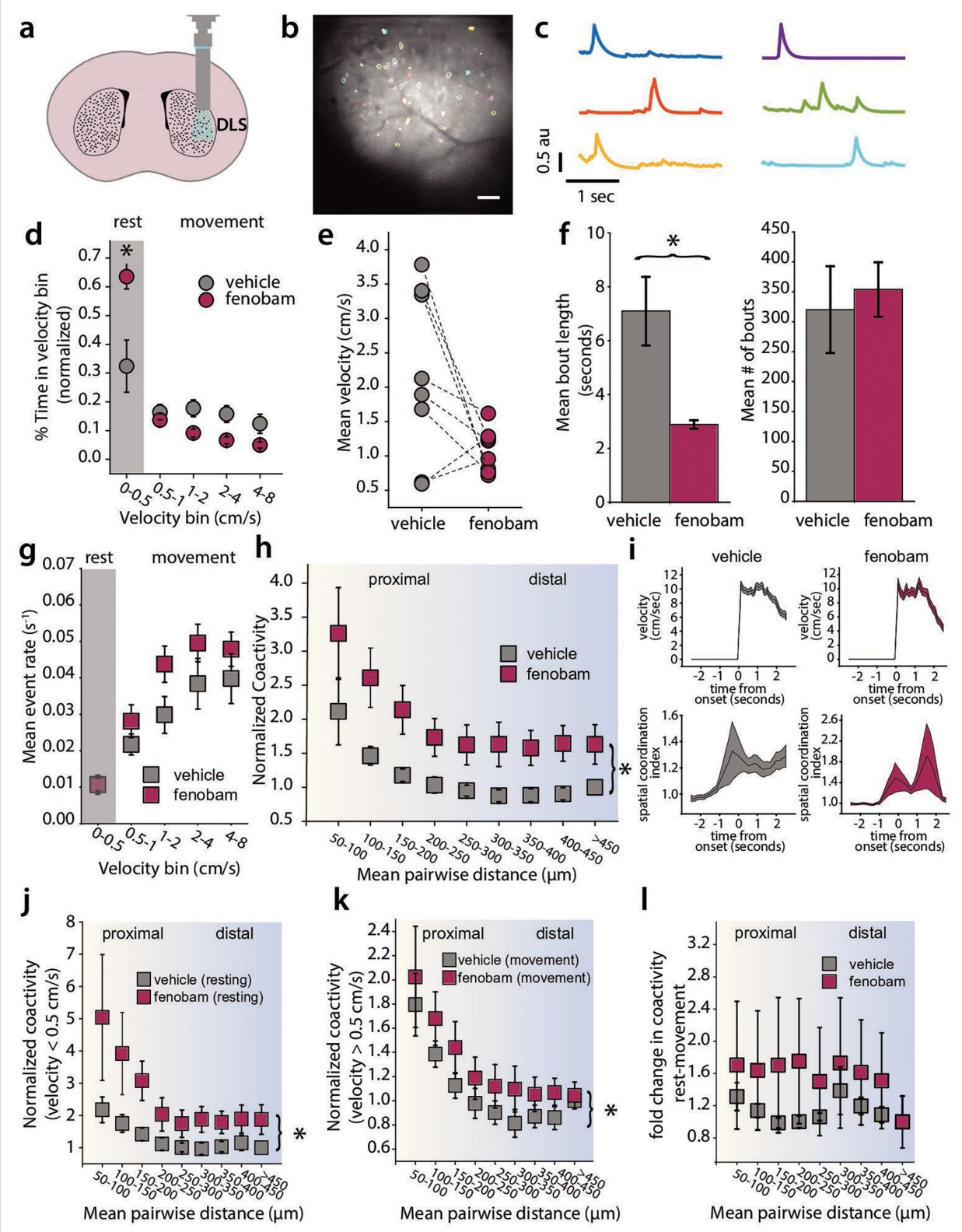

**Figure 2.** A negative allosteric modulator of mGluR5 increases coactivity among direct spiny-projection neuron (dSPN) pairs. (**a**) Cartoon representing implanted gradient refractive index (GRIN) lens and miniscope imaging in the dorsolateral striatum (DLS) (**b**) Examples of segmented cell outlines from the CNMF-E algorithm in a single mouse (scale 100 µm). (**c**) Example of Ca²⁺ traces from corresponding filled segmented cells shown in (**b**) (**d**) Binned velocity of animal. Fenobam increased the amount of time the animal spends at rest (Vehicle mean: 0.324±0.091, fenobam mean: 0.635±0.043, *p*=0.024,

*Figure 2 continued on next page*

*Figure 2 continued*

U=10, Mann-Whitney U test) (**e**) Mean velocity of animals following vehicle or fenobam treatment. The fenobam effect on mean velocity during the open field test is not significant (Mann-Whitney U test, $p=0.103$, U=48) (**f**) Mean bout length of animals during vehicle or fenobam. Fenobam reduces the mean bout length (Vehicle: 7.098 s±1.272, Fenobam: 2.886 s±0.160, Mann-Whitney U test, $p<0.001$, U=0), but does not affect the number of bouts (Mann-Whitney U test, $p=0.793$, U=35). (**g**) The average event rate of dSPNs at each movement velocity of the mouse. This relationship is not affected by fenobam administration (Linear Mixed Effects Model/LMM, $p=0.754$, $z=-0.313$, $\beta=-0.002$, SE = 0.006, 95% CI: [–0.014–0.010], N=8 mice). (**h**) Fenobam increases the pairwise coactivity of dSPNs when grouped into 50 μm bins (LMM, $p<0.000$, $z=-6.050$, $\beta=-0.828$, SE = 0.137, 95% CI: [-1.096–0.560], N=8 mice) (**i**) Following both vehicle and drug treatment, spatial coordination peaks during movement onset (Veh: $p<0.000$, z=12.21, Drug: $p<0.000$, z=10.1804) (**j**) Pairwise coactivity during rest with vehicle and fenobam treatment (velocity <0.5 cm*s$^{-1}$) (LMM, $p<0.000$, $z=-4.448$, $\beta=-1.292$, SE = 0.290, 95% CI: [-1.861–0.723], N=8 mice) (**k**) Pairwise coactivity during movement (velocity >0.5 cm*s$^{-1}$) with vehicle and fenobam (LMM, $p=0.008$, $z=-2.673$, $\beta=-0.220$, SE = 0.082, 95% CI: [-0.381–0.059], N=8 mice) (**l**) Fenobam does not affect the relative decrease in coactivity seen during movement. Error bars represent standard error of the mean, $p<0.05$ defines significance.

The online version of this article includes the following figure supplement(s) for figure 2:

**Figure supplement 1.** Fenobam does not affect Ca$^{2+}$ event length or amplitude.

treatment. While fenobam treatment increased dSPN co-activity during both rest and movement, its effects were most pronounced during periods of rest (*Figure 2j and k*). However, in analyzing the relative increase in coactivity between rest and movement, there was no significant difference between fenobam and vehicle-treated animals (*Figure 2l*). Taken together, these results suggest that fenobam may suppress movement by transitioning spatiotemporal dSPN ensemble dynamics towards a rest-associated state of high overall co-activity, without affecting the absolute rates of their activity. In this rest-associated state, overall neuronal activity is reduced but more correlated, resulting in a higher degree of co-activity.

## An mGluR5 positive allosteric modulator (PAM) increases movement and decreases SPN co-activity

Next, we sought to determine how promoting mGluR5 signaling affects motor output and SPN dynamics. We selected JNJ-46778212/VU0409551 (JNJ), an mGluR5 PAM with demonstrated in vivo efficacy and known pharmacokinetics in mice (*Conde-Ceide et al., 2015*). We administered JNJ (100 mg/kg) or vehicle via i.p. injection 30 min prior to measuring dSPN dynamics for 1 hr during locomotion in an open field arena. In contrast to fenobam, JNJ treatment decreased the amount of time the animal spent at rest (*Figure 3a*). However, there was no statistically significant effect on mean velocity (*Figure 3b*), number of bouts (*Figure 3c*), or bout length (*Figure 3d*). Like fenobam, JNJ did not affect the levels of dSPN activity when measured as a function of the animal's running speed (*Figure 3e*). Additionally, there was no effect on the amplitude or duration of Ca$^{2+}$ events (*Figure 3—figure supplement 1*). However, JNJ significantly decreased the co-activity among dSPN cell pairs measured across a range of Euclidean distances (*Figure 3f*). Again, as with fenobam, animals from both the vehicle and drug treatment groups showed significant increases in spatial coordination surrounding movement onset (*Figure 3g*). The effect on overall co-activity was most pronounced and statistically significant during periods of rest (*Figure 3i and j*). Therefore, JNJ treatment had effects that were qualitatively inverse to the effects of fenobam on locomotion and dSPN co-activity during periods of rest. These findings are consistent with the idea that JNJ promoted movement by inducing movement-associated dSPN ensemble dynamics by de-correlating their activity.

mGluR5 activation in the striatum results in the mobilization of eCBs, specifically 2-AG, in dSPNs (*Gerdeman et al., 2002*; *Kreitzer and Malenka, 2007*; *Marshall et al., 2018*; *Chevaleyre et al., 2006*). Therefore, we asked whether enhancing eCB levels by preventing 2-AG degradation using the monoacylglycerol lipase (MAGL) inhibitor JZL-184 affected the spatiotemporal dynamics of dSPN activity. We administered to mice JZL-184 (10 mg/kg i.p.) before recording dSPN Ca$^{2+}$ activity for 60 min in an open field arena. Interestingly, in contrast to the mGluR5 PAM, JZL-184 significantly reduced the rate of Ca$^{2+}$ events (*Figure 3k*) but did not affect the co-activity of dSPNs (*Figure 3l*). These results suggest that broadly enhancing eCB levels does not recapitulate the effects of potentiating mGluR5 signaling specifically.

Collectively, these results demonstrate that potentiating mGluR5 signaling shifts dSPN dynamics to a less correlated state, while inhibiting mGluR5 signaling enhances the correlated state of dSPNs. It is notable that spontaneous dSPN activity is less correlated during periods of movement than rest,

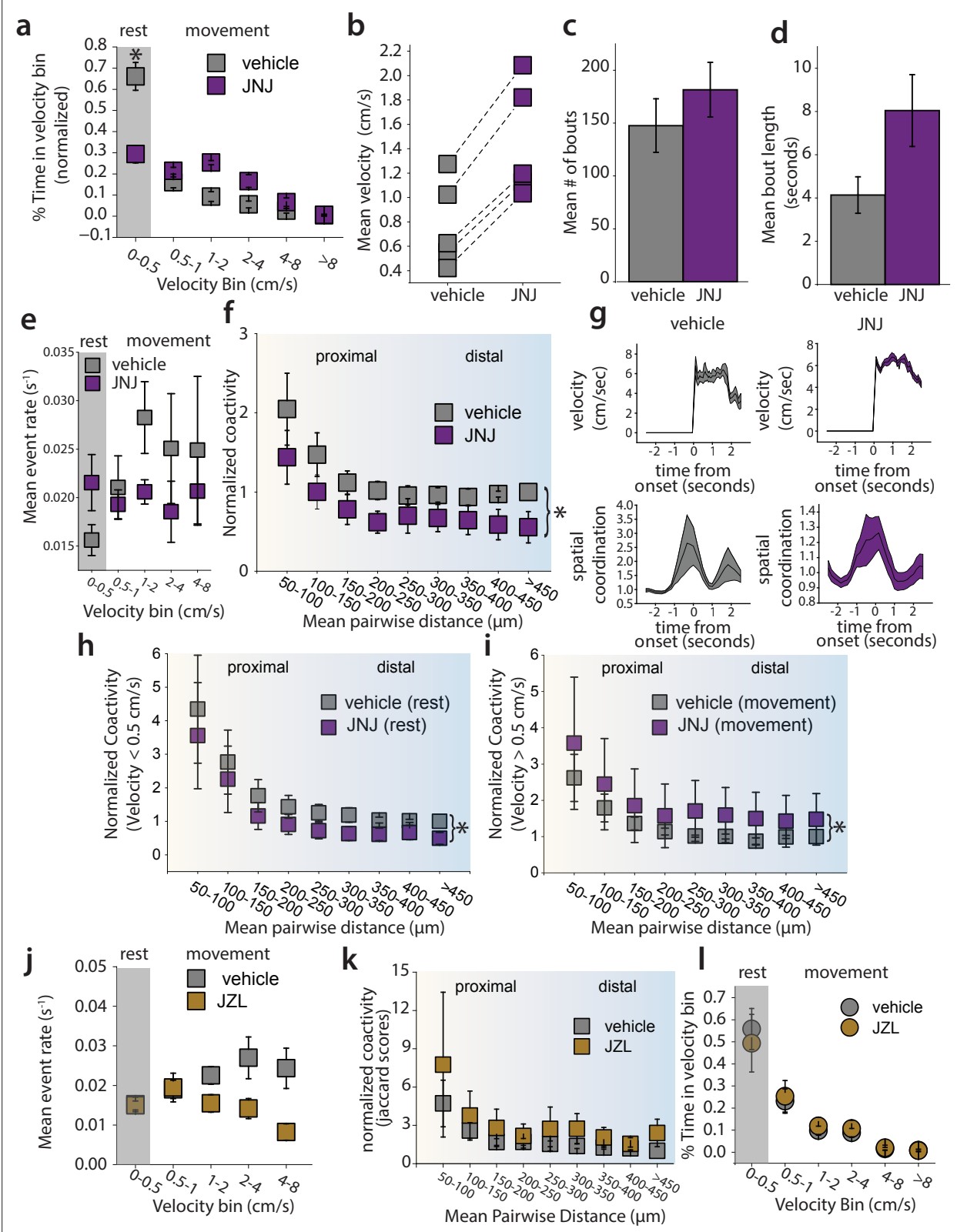

**Figure 3.** A positive allosteric modulator of mGluR5, JNJ-46778212, decreases clustered coactivity of direct spiny-projection neurons (dSPNs). (**a**) Time animal spent in each velocity bin and (**b**) Mean velocity of the animals during vehicle and after JNJ administration. JNJ decreases the amount of time the animal spends at rest (Vehicle mean: 0.661±0.066, JNJ mean: 0.294±0.041, p=0.012, U=25, Mann-Whitney U test), however, the change in mean velocity (**b**) is not significant (p=0.060, U=3, Mann-Whitney U test), nor is the mean bout length (**c**) during vehicle and JNJ (p=0.060, U=3, Mann-Whitney

*Figure 3 continued on next page*

*Figure 3 continued*

U test) (**d**) Comparison of mean number of bouts of each animal during vehicle and JNJ (p=0.403, U=8, Mann-Whitney U test). (**e**) Mean event rate of Ca$^{2+}$ transients in dSPNs during movement. JNJ administration does not significantly affect the event rate in dSPNs (LMM, p=0.058, z=1.893, $\beta$=0.010, SE = 0.005, 95% CI: [–0.000–0.020], N=4 mice) (**f**) Coactivity of pairs of dSPNs (50 μm bins) during total time in open field. Systemic JNJ administration reduces pairwise coactivity (LMM, p=0.003, z=2.964, $\beta$=−0.450, SE = 0.152, 95% CI: [–0.153–0.748], N=4 mice) (**g**) Spatial coordination increases at movement onset in both vehicle and drug groups (VEH: p<0.000, z=6.249, JNJ: p<0.000, z=9.111, Wilcoxon Signed Rank test) (**i**) Normalized co-activity of dSPNs during rest. JNJ produces a small but significant reduction in coactivity measured during rest (velocity <0.5 cm*s$^{-1}$) (LMM, p=0.030, z=2.165, $\beta$=0.618, SE = 0.285, 95% CI: [0.059–1.177], N=4 mice) (**j**) Normalized co-activity of dSPNs during movement (velocity >0.5 cm*s$^{-1}$). JNJ had a significant effect on dSPN clustered activity (LMM, p=0.027, z=−2.212, $\beta$=−1.694, SE = 0.766, 95% CI: [-3.195–0.193], N=4 mice).(**k**) Event rate in dSPNs during vehicle and after JZL administration. JZL-184 administration reduces the event rate (LMM, p=0.001, z=3.195, $\beta$=0.007, SE = 0.002, 95% CI: [0.003–0.012], N=5 mice) (**l**) Normalized co-activity of dSPNs against Euclidean distance. JZL had no effect on coactivity (LMM, p=0.065, z=−1.847, $\beta$=−1.204, SE = 0.652, 95% CI: [–2.481–0.074], N=5 mice) (**m**) Time animals spent in velocity bins. JZL did not affect time spent at rest (p=0.531, U=16, Mann-Whitney U test). Error bars represent standard error of the mean, p<0.05 defines significance.

The online version of this article includes the following figure supplement(s) for figure 3:

**Figure supplement 1.** JNJ does not affect Ca$^{2+}$ event length or amplitude.

and de-correlated dSPN activity has been shown to occur in hyperkinetic states (e.g. dyskinetic and hyperdopaminergic states; *Parker et al., 2018*; *Yun et al., 2023*). These observations suggest that inhibition of mGluR5 signaling suppresses movement by altering the spatiotemporal dynamics of dSPNs through synaptic mechanisms that alter their collective recruitment via excitatory afferents.

## mGluR5 reduces corticostriatal glutamatergic transmission at dSPN synapses in the dorsolateral striatum

Our experiments demonstrated changes in striatal dSPN network activity that correlate with changes in motor behavior mediated by pharmacologically modulating mGluR5. To determine whether modulating mGluR5, specifically in striatal dSPNs, mediates the observed effects on locomotor and striatal activity, we used a genetic approach to selectively manipulate mGluR5 in D1 receptor-expressing neurons. To do this, we crossed *Grm5$^{fl/fl}$* mice (*Xu et al., 2009*) to D1-Cre mice (*Gerfen et al., 2013*) to create conditional knockout mice (D1 cKO) mice. Further crossing with a Cre reporter allele (Ai9) (*Madisen et al., 2010*) enabled us to perform patch-clamp recordings from visually identified dSPNs in striatal slices to record corticostriatal EPSCs (*Figure 4a*). As we have previously demonstrated (*Marshall et al., 2018*), a 10-min application of the group 1 mGluR agonist DHPG (100 μM) caused a depression of EPSCs in WT recordings (*Figure 4b*). This effect was significantly diminished in recordings from cKO mice, demonstrating a disruption of mGluR5 signaling (*Figure 4c*).

A number of behavioral phenotypes have been described in mice with constitutive, germline *Grm5* deletion (*Xu et al., 2021*). Given our previous observations and the fact that SPN activity in the dorsal striatum is important for motor control, we asked whether selective ablation of *Grm5* in dSPNs affected locomotion and motor learning. Tracking distance traveled in the open field, we found that the D1 cKO mice had reduced spontaneous locomotion (*Figure 4d*) and displayed a reduction in total distance traveled (*Figure 4e*). We also found the D1 cKO mice had diminished digging behavior in a novel environment (*Figure 4f*). Taken together, these results demonstrate that selectively disrupting mGluR5 signaling in dSPNs reduces spontaneous movement in novel environments.

To measure locomotion in the home cage setting, we introduced running wheels and monitored activity over 7 days. During the first 3 days, D1 cKO mice demonstrated significantly reduced voluntary running (*Figure 4h and i*). However, both control and D1 cKO mice increased the time spent running over the course of 7 days such that the difference between the groups was not significant by day 7 (*Figure 4i*).

Finally, on an accelerating rotarod, D1 cKO mice showed a consistently reduced latency to fall over the course of 12 training trials across three days (*Figure 4g*). Despite this deficit, D1 cKO mice did improve over the course of training, indicating that they were capable of learning the task. Thus, the D1 cKO mice exhibited a primary deficit in spontaneous motor activity with only minor deficits in the learned motor tasks of running on a wheel or forced locomotion on the rotarod.

Because of the clear motor phenotype in D1 cKO mice, we determined whether there were any changes in the intrinsic or synaptic properties of dSPNs in these mice. Whole-cell current-clamp recordings from visually identified dSPNs expressing tdTomato found no difference in the voltage

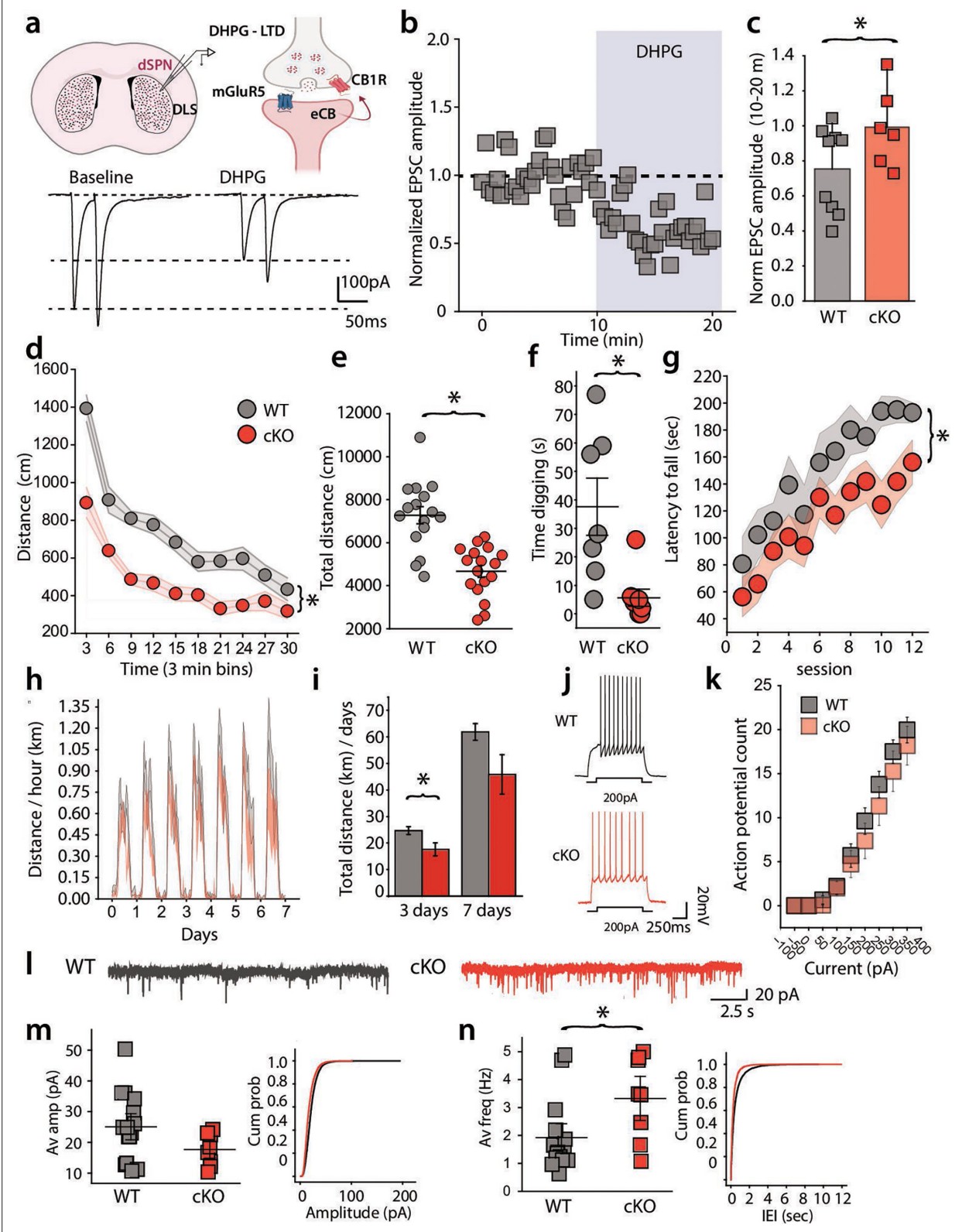

**Figure 4.** mGluR5 conditional knockout (cKO) in Drd1a-expressing neurons reduces spontaneous motor behaviors and affects synaptic properties of direct spiny-projection neurons (dSPNs). (**a**) (Top) Cartoon representation of the role of mGluR5 in mobilizing endocannabinoids (eCBs) at corticostriatal synapses. (Bottom) Representative EPSCs recorded prior to and during DHPG in wild-type (WT) mice (**b**) Example time course of corticostriatal EPSCs recorded in dSPNs during application of 100 μM DHPG in slice from WT mice. (**c**) Grouped data of DHPG depression. D1 cKO mice lack DHPG-induced

*Figure 4 continued on next page*

*Figure 4 continued*

synaptic depression (cKO normalized EPSC amplitude: 0.754±0.082, Wilcoxon Signed Rank Test, *p*=0.018, W=2; WT normalized amplitude: 0.992±0.992, Wilcoxon Signed Rank Test, *p*=0.834, W=9). (**d**) Time course of activity in open field and (**e**) Total distance traveled by WT and cKO mice. D1 cKO mice have reduced activity in the open field, measured as distance traveled over time and total distance traveled during a 30 min open-field session (Mann-Whitney U Test, U=245, *p*<0.001, N=16 WT, 17 cKO). (**f**) Digging time in WT and cKO mice. D1 cKO mice spend less time digging when placed into a novel home cage (Mann-Whitney U Test, U=51.5, *p*=0.008, N=7 WT, 8 cKO). (**g**) Latency to fall in the accelerating rotarod test. dSPN cKO mice have consistently reduced latency during the accelerating rotarod test over 3 days, with four sessions per day (*p*=0.024, F=5.709, Two-way repeated measures ANOVA, N=12 WT, 17 cKO). (**h**) Running distance per hour on wireless wheels placed in home cage over 7 days. (**i**) Total distance run per day during the first 3 days and over 7 days. D1 cKO mice ran significantly less during the first 3 days of testing (Mann-Whitney U Test, U=173, *p*=0.049) (**j**) Example traces of voltage response to 200 pA current injection in WT and cKO mice. (**k**) There is not a significant difference in the input-output curve for current injection and AP firing between genotypes (LMM, *p*=0.056, z=1.913, *β*=1.197, SE = 0.626, 95% CI: [−0.029–2.424], N=26 cells, WT, N=22 cells, cKO). (**l**) Example mEPSC recordings in dSPN from WT and D1 cKO mice (**m**) Average amplitude of mEPSCs and cumulative distribution of amplitudes in WT and D1 cKO mice. There is no difference in amplitudes in recordings from the two genotypes (WT AMP: 21.04±1.71 pA, N=15, cKO AMP: 16.66±1.12, N=8, Mann-Whitney U Test, U=86.5, *p*=0.093) (**n**) Average mEPSC frequency and cumulative interevent intervals in WT and D1 cKO animals. D1 cKO mice have increased mEPSC frequency (WT Hz: 1.920±0.3319 pA, N=15, cKO Hz: 3.325±0.525, N=8, Mann-Whitney U Test, U=27.5, *p*=0.039). Error bars represent standard error of the mean, p<0.05 defines significance.

response to increasing current injection, suggesting no alteration in the intrinsic excitability of dSPNs in D1 cKO mice (*Figure 4j and k*). In recordings of mEPSCs that reflect all excitatory synapses (corticostriatal and thalamostriatal), we found that the amplitudes of mEPSCs were unchanged, but their frequency was significantly increased in dSPNs from D1 cKO mice (*Figure 4l–m*), indicating increased release probability when mGluR5 expression is disrupted.

## Spatiotemporal activity of dSPNs is disrupted in D1 cKO mice

To measure the in vivo dynamics of dSPNs in D1 cKO mice, we implanted GRIN lenses into the DLS of control and D1 cKO mice and analyzed dSPN Ca$^{2+}$ activity during 20 min of open field exploration (*Figure 5a–e*). We performed control imaging experiments in mice expressing Cre in dSPNs but with the wild-type *Grm5* allele (D1 cre; *Grm5* +/+). Consistent with our previous results, the event rate of dSPN Ca$^{2+}$ transients increased with locomotor speed in both experimental and control mice (*Figure 5d*). Selective mGluR5 deletion in *Drd1a*-expressing neurons did not alter the rate of dSPN Ca$^{2+}$ events compared to control mice (*Figure 5d*). cKO mice and control animals also showed similar distributions of event amplitudes and durations (*Figure 5—figure supplement 1*). Instead, the normalized coactivity of dSPNs (analyzed across 50 µm bins of pairwise cell distances) was significantly increased in D1R cKO mice, similar to the effect of the mGluR5 NAM fenobam (*Figure 5e*). Additionally, D1R cKO mice and control mice showed a transient peak in coactivity surrounding motion onset (*Figure 5f*). When measured over all frames, the coactivity in both control and cKO mice increased during continuous periods of rest compared to periods of continuous movement; however,, there was no difference in this effect between groups (*Figure 5g*). Therefore, targeted genetic deletion of mGluR5 in dSPNs had qualitatively similar effects as systemic administration of the mGluR5 NAM on co-activity of dSPNs.

Finally, because both dSPNs and iSPNs exhibit spatially clustered activity in vivo, and because they are synaptically connected through recurrent collaterals (*Taverna et al., 2008*), it is possible that disruption of mGluR5 signaling in dSPNs may also affect the levels and spatiotemporal dynamics of activity in intermingled iSPNs. To address this question, we used a viral 'Cre-off' strategy to express GCaMP6 in Cre-negative, iSPNs of *Drd1a*-Cre; *Grm5*$^{fl/fl}$ mice (*Saunders et al., 2012*). We found that the deletion of mGluR5 in dSPNs had no effects on either the levels of activity in iSPNs (*Figure 5h*) or their spatiotemporal clustering (*Figure 5i*). Therefore, the alteration in SPN dynamics in D1 cKO mice appears to be cell-autonomous, as specific targeting of mGluR5 in dSPNs did not affect iSPN activity in D1 cKO mice.

Together, these results demonstrate that the manipulation of mGluR5 signaling, through pharmacological or targeted genetic means, leads to changes in the spatiotemporal patterns of activity in dSPNs with minimal effects on their overall levels of activity. Notably, these spatiotemporal dynamics varied with locomotor speed under normal conditions, and manipulations of mGluR5 signaling induced changes in these dynamics that correlated with locomotor state. These findings suggest that these two facets of activity—levels and spatiotemporal patterning of activity—are separable and

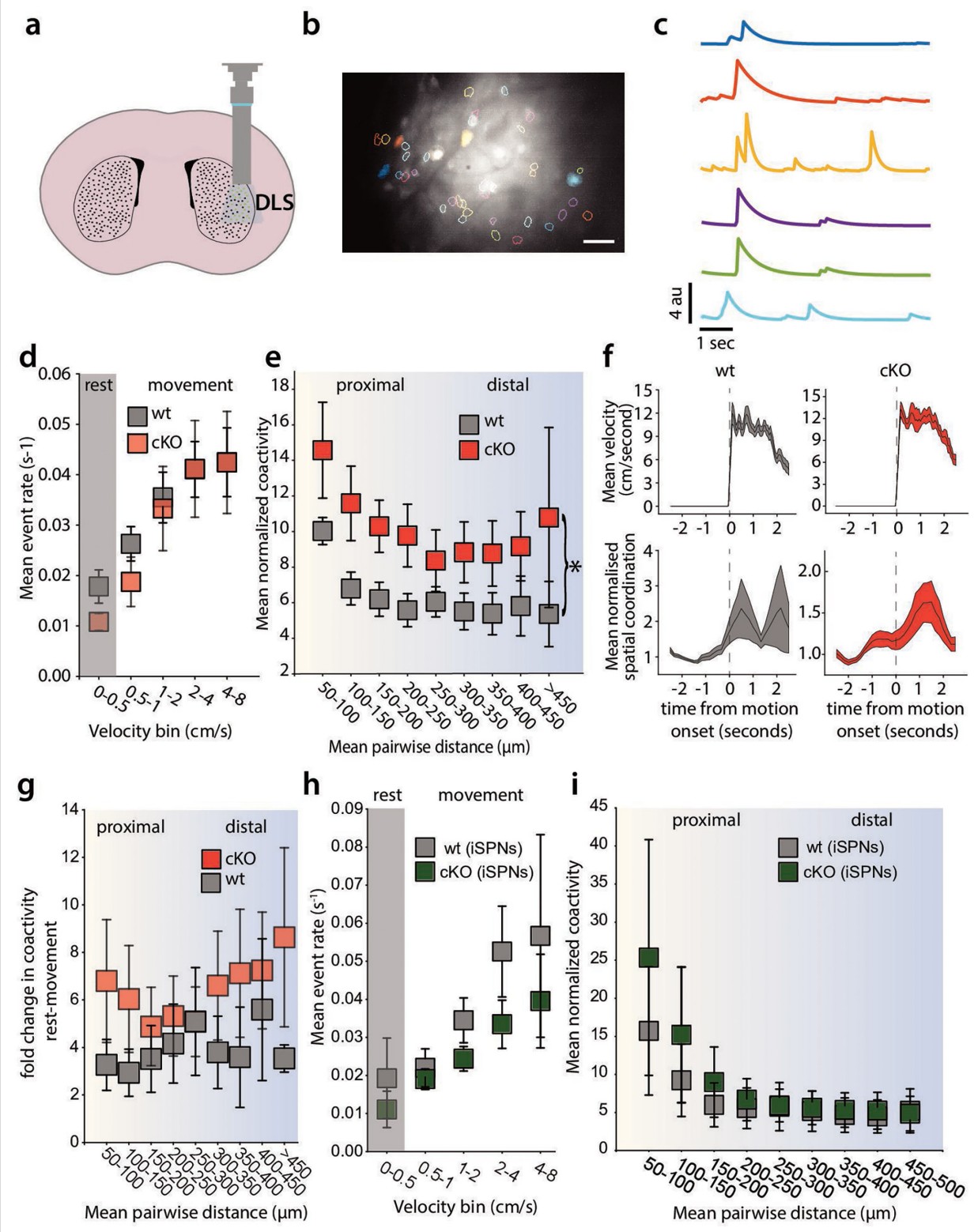

**Figure 5.** D1 conditional knockout (cKO) mice show cell-autonomous increase in clustered coactivity in direct spiny-projection neurons (dSPNs).
(**a**) Cartoon representation of miniscope imaging in the dorsolateral striatum (DLS). (**b**) Example segmented image from cKO mice (scale 100 μm).
(**c**) Example Ca²⁺ activity traces from identified dSPNs in cKO mice, filled cells in (**b**) (**d**). Mean event rate during movement in wild-type (WT) and cKO
animals. The event rate in dSPN cKO mice is unchanged from WT (LMM, $p$=0.651, z=0.452, $\beta$=0.003, SE = 0.007, 95% CI: [–0.011–0.017], N=9 wt, 6
cKO). (**e**) Coactivity of pairs of dSPNs (50 μm bins) in WT and cKO animals. D1 cKO mice have increased coactivity across all pairwise distances (MLM,

*Figure 5 continued on next page*

*Figure 5 continued*

$p$=0.021, z=2.305, $\beta$=3.989, SE = 1.731, 95% CI: [0.597–7.382], N=9 wt, 6 cKO) (**f**) Spatial coordination peaks at movement onset in both wt and cko mice (WT: $p$<0.000, z=7.301, KO: $p$<0.000, z=6.700, Wilcoxon Signed Rank test). (**g**) The relative increase in coactivity during rest is not statistically different between cKO and wt mice (LMM, $p$=0.552, z=0.595, $\beta$=1.391, SE = 2.339, 95% CI: [–3.193–5.976]). (**h**) indirect spiny-projection neurons (iSPN) event rate in D1 cKO and WT animals is unchanged (LMM, $p$=0.310, z=1.016, $\beta$=0.017, SE = 0.016, 95% CI: [–0.015–0.048], N=3 wt, 3 cKO). (**i**) Clustered coactivity of iSPNs in WT and D1 mGluR5 cKO animals is unaffected (LMM, $p$=0.691, z=0.398, $\beta$=2.368, SE = 5.953, 95% CI: [–9.300–14.035], N=3 wt, 3 cKO). Error bars represent standard error of the mean, $p$<0.05 defines significance.

The online version of this article includes the following figure supplement(s) for figure 5:

**Figure supplement 1.** $Ca^{2+}$ event lengths and amplitudes do not differ between wt and cKO mice.

provide important insights into the role of spatiotemporally coordinated activity in the generation of movement.

## Discussion

The rate-model of basal ganglia function posits that activity in dSPNs and iSPNs have distinct and opposing effects on movement. This framework is primarily based on anatomical evidence from the distinct projection targets of dSPNs and iSPNs and electrophysiological recordings in their downstream projection targets under conditions modeling Parkinsonism (*Nelson and Kreitzer, 2014*). These target nuclei inhibit motor control nuclei in the hindbrain mesencephalic locomotor region and thalamic regions that project to the motor cortex (*McElvain et al., 2021*; *Parent and de Bellefeuille, 1982*). In this context, dSPN activation is predicted to increase activity in these motor control regions to facilitate movement, while iSPN activation is predicted to decrease activity in these same regions to inhibit movement.

However, this classical model is not supported by recent in vivo imaging studies using fiber photometry or endoscopy to record $Ca^{2+}$ activity in genetically specified dSPNs or iSPNs. In contrast to the predictions of the rate model, dSPNs and iSPNs are indistinguishably responsive to increases and decreases in movement (*Cui et al., 2013*). Intriguingly, in vivo endoscopy has shown that overlapping clusters of dSPNs and iSPNs co-activate during movement, with small but statistically significant correlations between the activation of different clusters during different movement types (*Barbera et al., 2016*). These findings suggest an alternative model where, in contrast to the relative level of activity in each SPN type, the degree of co-activity among SPNs and ensemble identity may be important for specifying proper motor output.

Despite evidence that SPNs may influence movement via the coordination of their activity, the biological mechanisms responsible for these coordinated patterns are not known. Seminal in vitro electrophysiology studies demonstrated that SPNs fire action potentials coincident with large excitatory inputs that shift their resting membrane potential to transiently stable, depolarized 'up' states (*Wilson and Groves, 1981*; *Wilson and Kawaguchi, 1996*). These excitatory inputs arise from the cortex and thalamus. Our results suggest that the modulation of the synaptic weight, particularly at corticostriatal synapses where several forms of plasticity require mGluR5 receptors, is a crucial mechanism constraining the patterns of dSPN co-activity in states associated with movement or rest.

### Spontaneous motor output correlates with the level of coactivity in dSPNs

In the current work, we used pharmacological and genetic manipulations to examine how activity in dSPNs varies with movement while targeting the receptor controlling the dominant form of synaptic plasticity in the DLS. Systemic administration of the mGluR5 negative allosteric modulator, fenobam, led to significant increases in co-activity among dSPNs and reduced locomotor activity. By contrast, systemic administration of the mGluR5 positive allosteric modulator JNJ decreased dSPN co-activity and increased locomotion. Notably, neither pharmacological manipulation of mGluR5 led to significant changes in the rate of dSPN $Ca^{2+}$ events. Our results suggest a model whereby either potentiating or inhibiting mGluR5 signaling alters the recruitment of dSPNs such that the overall activity level is preserved but that the patterns of their co-activity change in a manner that corresponds to motor output. This is further confirmed by our data from animals with targeted deletion of mGluR5 in dSPNs, which similarly showed no change in event rate but an increase in co-activity.

Our results are consistent with prior analysis demonstrating that the co-activity among pairs of dSPNs is higher at rest than during movement (*Yun et al., 2023*). Here, we found that pharmacological manipulations that alter these clustered co-activity states correspondingly alter the levels of spontaneous motor output. Specifically, shifting the dSPN network to a level of higher co-activity, as with fenobam treatment or mGluR5 deletion in dSPNs, leads to a reduction in the amount of time animals spent moving. By contrast, potentiating mGluR5 receptors with systemic JNJ administration led to an increase in time spent moving and a corresponding decrease in the dSPN co-activity.

Our ex vivo electrophysiology data from cKO animals further supports these conclusions, as disrupting endogenous mGluR5 signaling in dSPNs does not result in changes in the excitability of neurons. Instead, genetic deletion of mGluR5 altered the synaptic properties of dSPNs, likely an adaptive response consistent with the established role of mGluR5 in regulating corticostriatal synaptic weights. These findings suggest that these selective changes in synaptic plasticity may underlie the altered spatiotemporal patterning of dSPN activity without changes in their overall levels of activity observed in vivo.

It is important to note that, in our pharmacological experiments, we observed that inhibition of mGluR5 with systemically administered fenobam increased the co-activity among active neuron pairs across all distance bins we measured. That is, co-activity was increased in both proximal and distal neuron pairs, and this increased co-activity correlated with decreased ambulatory behavior. Administration of an mGluR5 PAM decreased co-activity when measured across all binned distances of neuron pairs, and this occurred in coincidence with increased ambulatory behavior. It has previously been demonstrated (*Parker et al., 2018*), and we replicated here, that surrounding movement onsets, there is an increase in the extent to which proximal neuron pairs are statistically more likely to be co-active than distal neuron pairs. Using a previously established metric for quantifying this increase, the spatial correlation index, we did not resolve any significant changes in the onset-related changes in this metric following either genetic or pharmacological alterations of mGluR5 activity. Our central finding, observed with both pharmacological and genetic manipulations, is that manipulating mGluR5 signaling alters overall dSPN co-activity, irrespective of cell proximity, in a manner that correlates with changes in motor output. We hypothesize that these changes are likely due to alterations in synaptic weight at dSPN synapses that occur following mGluR5 manipulation.

One way in which mGluR5 may modify synaptic weight is through long-term plasticity. In vitro studies have demonstrated that mGluR5 activation occurs following pairing of postsynaptic depolarization with moderate to high frequency excitatory synaptic input (*Gerdeman et al., 2002*). mGluR5 activation can result in the long-term depression (LTD) of synaptic transmission or a more transient depression in release probability (*Kreitzer and Malenka, 2007*; *Marshall et al., 2018*; *Shen et al., 2015*). If rest-associated ensembles of dSPNs are substantially different in composition from movement and/or movement-onset ensembles, then our data suggest that synapses onto dSPNs may have a lower synaptic weight during spontaneous behavior than movement or movement onset ensembles, potentially due to eCB release. Inhibiting mGluR5, either through pharmacology or genetics, may increase this synaptic weight, thereby promoting activation of rest ensembles and decreasing motor output. This input that primarily governs the activation of movement-related ensembles could either be a separate population of cortical inputs with higher synaptic weight or thalamostriatal inputs more resilient to mGluR5 manipulations (*Wu et al., 2015*).

Our results are an important step in understanding how the regulation of synaptic weights by mGluR5 may act on the coordinated activity of dSPNs in vivo to regulate behavior.

## Global enhancement of eCB levels does not alter clustered co-activity

The lack of change in spatiotemporal coordination following systemic administration of the 2-AG degradation inhibitor JZL-184 suggests that global enhancement of eCB levels does not produce a similar outcome as modulating mGluR5 signaling. However, in contrast to selective mGluR5 modulation, we did observe a reduction in overall dSPN event rates with JZL. One possibility is that a degree of synaptic specificity is required to affect clustered activation of dSPNs. Thus, JZL treatment may have exerted such broad effects on the circuit that it resulted in a different outcome for neural clustering and motor output than that observed following treatment with the allosteric modulators of mGluR5 receptors that modify the weight of specific active synapses.

These results support a model whereby the level of co-activity among specific groups of dSPNs is potentially important for controlling motor output. Therefore, globally altering the synaptic weights of all neurons may overshadow the results of synaptic alterations in specific subsets of SPNs. This idea is consistent with the notion that individual dSPNs may be precisely tuned to have specific roles in regulating motor output, such that altering the activity of one dSPN subgroup would not have the same effect as altering the activity of a different subgroup. This interpretation is also at odds with the classical rate-based models of SPN activity.

## Role of non-dSPN mGluR5 receptors in regulating movement

There are notable differences in the design of our experiments using conditional deletion of mGluR5 from dSPNs and systemic administration of a negative allosteric modulator of mGluR5. Following fenobam administration, we observed a significant increase in co-activity when animals were at rest and a smaller increase during movement; however,, the relative increase in co-activity between rest and movement did not differ significantly from controls in either cKO animals or following fenobam treatment. Therefore, in both cases, inhibition of mGluR5 may promote the activation of rest ensembles and thereby inhibit motor output.

Within the striatum alone, mGluR5 is expressed in iSPNs and interneurons, and modulation of these cell types by systemically administered compounds could produce changes in behavior and striatal circuit function. dSPNs receive GABAergic input from both iSPNs as well as striatal interneurons. Of particular note, parvalbumin-positive fast-spiking interneurons also receive excitatory input from the cortex and provide feedforward inhibition onto both subtypes of SPNs, and genetic disruption of mGluR5 in parvalbumin interneurons increases striatum-dependent repetitive behaviors (*Barnes et al., 2015*). The modulation of synaptic strength at these connections by fenobam and JNJ could also partly account for the effects of these compounds on dSPN activity and behavior when systemically administered, which would not occur in cKO animals.

Collateral inhibition between SPNs is another potential source for modulation of co-activity. In vitro electrophysiology has shown that dSPNs are more likely to form synapses onto other dSPNs, while iSPNs form synapses onto neurons of both pathways with equal probability (*Taverna et al., 2008*). However, the strength of collateral synapses formed by dSPNs appears to be smaller than that of synapses formed by iSPNs. Our analysis of iSPN activity in cKO mice suggests that iSPN circuit properties are largely normal when mGluR5 is genetically disrupted in dSPNs, perhaps reflecting the less substantial modulation of iSPN circuitry by dSPNs reported in the striatum, at least during spontaneous behavior. Whether this relationship holds across additional behavioral states remains to be addressed. Additionally, whether disruption of iSPN networks affects dSPN networks, perhaps due to stronger unidirectional iSPN-dSPN connectivity, is a key question for further research.

## Translation implications of dSPN spatiotemporal dynamics

The relationship of mGluR5 pharmacology to dSPN spatial coordination and motor output is clinically relevant to the treatment of motor disorders. Prior studies examining the spatial coordination of SPNs in Parkinsonian mice demonstrated that, following dopamine depletion to model a Parkinsonian state, dSPN activity was reduced but maintained spatial coordination (*Parker et al., 2018*). However, when levodopa-induced dyskinesia (LID) was modelled in these animals, dSPNs lost spatial coordination. In the current work, we show that pharmacological inhibition of mGluR5 increases spatial coordination. It is tempting to speculate, therefore, that in the case of LID, the therapeutic efficacy of mGluR5 NAMs may involve regulation of the spatially coordinated co-activity of dSPNs. Additionally, further examination of how spatially coordinated SPN activity is disrupted in motor diseases involving basal ganglia pathology may be a promising avenue for future therapeutic development.

In summary, our data demonstrate that the modulation of signaling that affects synaptic properties of excitatory synapses onto dSPNs is a key mediator of the normal patterns of SPN co-activity that typify the striatum's neural ensemble dynamics in relation to spontaneous movement. These results provide further support for models where the local synchronization of SPNs (and the basal ganglia circuit in general) is a stronger determinant of motor output than their overall levels of activity. Our

results also provide a framework for future lines of inquiry to address how striatal synaptic properties shape circuit function and striatum-dependent behavior.

# Methods

## Key resources table

| Reagent type (species) or resource | Designation | Source or reference | Identifiers | Additional information |
|---|---|---|---|---|
| Strain, strain background (*Mus musculus*) | Drd1a-Cre Ey262 | MMRRC repository | RRID:MMRRC_030989-UCD; D1-Cre\|D1R Cre\|EY262\|Tg(Drd1a-cre)EY262Gsat\|Tg(Drd1a-cre)262Gsat | D1 cre mice |
| Strain, strain background (*Mus musculus*) | mGluR5$^{loxP/loxP}$ | https://doi.org/10.1523/JNEUROSCI.5716-08.2009 | RRID:IMSR_JAX:028626; mGluR5$^{loxP/loxP}$ | floxed mGluR5 mice |
| Strain, strain background (*Mus musculus*) | Ai9 | The Jackson Laboratory | RRID:IMSR_JAX:007909; Strain #: 007909 | Ai9 reporter mice |
| Other | pAAV.Syn.Flex.GCaMP6f.WPRE.SV40 | Addgene | RRID:Addgene_100833; 100833-AAV9 | AAV virus (AAV9) |
| Other | pAAV.Syn_hGH.DO_FAS.GCaMP6f.WPRE.bGHpA | This paper | pAAV.Syn_hGH.DO_FAS.GCaMP6f.WPRE.bGHpA, available from authors | AAV virus (AAV9) |
| Chemical compound | Silk Fibroin Solution, 50 mg/ml | Advanced Biomatrix | Cat # 5154–20 ML | Silk fibroin for coating GRIN lens |
| Other | 1 mm GRIN lens | Inscopix | 1050–004595 | GRIN lenses 1 mm diameter, 4 mm length. |
| Other | 1.8 mm GRIN lens | Edmund Optics | 64–519 | 1.8 mm Dia, 670 nm DWL, 0.0 mm WD, Uncoated, GRIN Lens |
| Other | Metabond | Parkell | SKU: S380 | C&B Metabond Quick Adhesive Cement System |
| Other | Data Acquisition System (DAQ) | LABmaker | Data Acquisition System (DAQ) | Miniscope DAQ for v3 |
| Other | UCLA v3 miniscope body components | https://miniscopeparts.com/ | V3 Complete Set | UCLA v3 miniscope components |
| Other | UCLA v3 complete list of optical and hardware components | http://miniscope.org/ | Other | Complete list found under "V3 Master Parts List" |
| Other | Miniscope v3 baseplate | https://miniscopeparts.com/ | V3.2 Base Plate Complete with magnets and 0-80 set screw cup point | |
| Other | Miniscope V4 - Complete set of components | LABMaker | Other | complete kit for assembling v4 scope |
| Other | UCLA v4 miniscope components | https://miniscopeparts.com/ | Other | UCLA v4 miniscope components |
| other | V4 Variant 1 Base Plate w/ 0-80 set screw cup point | https://miniscopeparts.com/ | Other | UCLA v4 Baseplate |

*Continued on next page*

*Continued*

| Reagent type (species) or resource | Designation | Source or reference | Identifiers | Additional information |
|---|---|---|---|---|
| Other | Miniscope V4 - Data Acquisition System (DAQ) | LABmaker | Other | UCLA v4 DAQ |
| Software, algorithm | Miniscope QT Software | https://github.com/Aharoni-Lab/Miniscope-DAQ-QT-Software | | |
| Software, algorithm | Denoising code for v4 scope | https://github.com/Aharoni-Lab/Miniscope-v4/tree/master/Miniscope-v4-Denoising-Notebook | | |
| Software, algorithm | CaImAn (normcorr) | https://github.com/flatironinstitute/CaImAn; *Gunn et al., 2026*; *Giovannucci et al., 2019* | | normcorr algorithm for motion correction |
| Software, algorithm | CNMF_E | https://github.com/zhoupc/CNMF_E; *Zhou and Heins, 2024*; *Zhou et al., 2018*; *Pnevmatikakis et al., 2016* | | CNMF_E algorithm for calcium signal extraction |
| Software, algorithm | ezTrack | https://github.com/denisecailab/ezTrack; *Pennington and Cai, 2021*; *Pennington et al., 2019* | | for tracking mouse position |
| Software, algorithm | statsmodels | https://www.statsmodels.org | RRID:SCR_016074 | Python package used for statistical analysis in paper |
| Software, algorithm | MATLAB | https://www.mathworks.com/products/matlab.html | RRID:SCR_001622 | |
| Software, algorithm | ANACONDA python distribution | https://anaconda.org/ | RRID:SCR_025572 | |
| Antibody | Mouse anti-GFP, monoclonal | Millipore | RRID:AB_94936; MAB3580 | (1:10,000) |

## Mice

Animals were group housed with 14 hr:10 hr Light/Dark cycle and food and water were provided ad libitum. Mice of both sexes, between 4–12 months old, were used and all experiments were performed in accordance with procedures approved by the Northwestern University IACUC. To express GCaMP6f in dSPNs, we used mice expressing Cre recombinase under the control of the *Drd1a* promoter (D1cre, Drd1a-Cre Ey262) (*Gong et al., 2007*). A Cre-dependent GCaMP6f under the control of the synapsin promoter (pAAV.Syn.Flex.GCaMP6f.WPRE.SV40, 100833-AAV9 from Addgene) was used for delivery of the construct. To express GCaMP6f in iSPNs in the D1cre mice, we generated a novel AAV vector with GCaMP6f under control of the synapsin promoter and downstream of a DO (Double-floxed Open reading frame) which prevents expression of the construct in cells expressing Cre recombinase (*Saunders et al., 2012*). This construct (pAAV.Syn_hGH.DO_FAS.GCaMP6f.WPRE.bGHpA) was packaged

into an AAV9 virus (Virovek, Houston, TX). For in vitro patch clamp electrophysiology experiments, we further crossed the D1Cre line to mice expressing a Cre-dependent tdTomato reporter inserted in the *Rosa26*$^{Ai9}$ locus (Ai9), in order to identify dSPNs in the DLS (*Madisen et al., 2010*). For genetic deletion of mGluR5, we used floxed mGluR5 mice (Grm5$^{loxP/loxP}$) (*Xu et al., 2009*) crossed to either D1cre mice (D1cre; mGluR5$^{f/f}$, or D1 cKO), for in vivo imaging experiments, or D1 cre mice crossed with Ai9 mice (D1cre; mGluR5$^{f/f}$; Ai9$^{+/+}$) for patch clamp experiments. All mice were maintained on the C57BL6/J background strain.

## Behavioral analysis
### Motor learning on the accelerating rotarod
Mice of both sexes were placed on the rotarod and the initial revolution speed set to 4 rpm for 120 s. The rotarod then accelerated at a rate of 0.12 rpm·s$^{-1}$ for 300 s. Mice were trained in three trials per day with a 30 min inter-trial interval for four consecutive days (12 trials total).

### Open field
The dimensions of the square open field arena (in cm) were 60 l×60 w×30 h. Animals were placed in the middle of the arena at the start of the test and remained in the arena for 30 min during which time their position was tracked using an overhead camera and analyzed using software (Limelight; Actimetrics).

### Digging behavior
Mice were placed in a novel cage containing wood chip bedding (3 cm deep) in a brightly lit, open area. Animals remained in the testing cage for 15 min. Mouse digging time was manually scored using a stopwatch.

### Voluntary wheel running
Mice of either sex were single-housed with wireless running wheels (MedAssociates) for at least 7 days. Total distance run was quantified for each day.

### Virus injection
For mice that received a viral injection (pAAV.Syn.Flex.GCaMP6f.WPRE.SV40, 100833-AAV9 from Addgene) prior to GRIN lens implant, we anesthetized mice with isoflurane (2% in O₂) and stereotaxically injected virus at a rate of 250 nl min$^{-1}$ into the DLS using a microsyringe with a 28-gauge beveled-tip needle (Hamilton Company, Reno, NV). Stock titer virus was diluted 1:10 with sterile PBS. The coordinates for injection were: A/*P*+/-0.05 mm, M/L+/-1.8 mm, D/V –2.4 mm (from dura). For all DV coordinates, we went 0.1 mm past the injection target and then withdrew the syringe back to the target for the injection. After each injection, we left the syringe in place for 10 min and then slowly withdrew the syringe. We then sutured the scalp, injected analgesic (Buprenorphine SR, 1 mg kg$^{-1}$), as well as meloxicam (2x/day for 48 hr, 5 mg·kg$^{-1}$), and allowed the mice to recover for at least 14 days.

### Silk fibroin coating of GRIN lenses
For animals in which we implanted a GRIN lens coated with a fibroin:AAV mixture, we autoclaved both GRIN lenses and a small aliquot of 5% W/V silk fibroin solution prior to coating. Then, the sterile fibroin solution was mixed 1:1 with stock titer AAV (pAAV.Syn.Flex.GCaMP6f.WPRE.SV40, 100833-AAV9 from Addgene). A microsyringe with a 23-gauge flat tip needle (Hamilton Company, Reno, NV) with a manual stereotaxic injector (Stoelting) was used to apply a 1 μL droplet of the mixture to the tip of the GRIN lens in a biological safety cabinet. The mixture was allowed to dry slightly for 5 min with the droplet in contact with both the GRIN lens surface and the syringe needle tip before the syringe needle was removed; this prevented the mixture from coating the side of the lens. Once the needle was removed, the droplet was allowed to fully dry to the surface of the lens for at least 1 hr before implanting. Lenses were stored at 4°C for implanting either the same day or within approximately 24 hr.

## Surgical implantation of GRIN lens and viral transduction

Mice were anesthetized with isoflurane in $O_2$ (induction 4%; maintenance 2–3%). Following verification of anesthesia, hair was trimmed from the incision site, the scalp cleaned with three alternating swabs of betadine and 70% ethanol, and an incision made through the scalp to uncover the skull. The skull surface was cleaned and verified to be level by measuring the coordinates of bregma, lambda, and two sites +/-2 mm lateral to the midline. The M/L and A/P coordinates for implantation of the GRIN lens (Inscopix for v4 lenses, or Edmund Optics for v3 lenses) were marked on the skull (A/P+/-0.05 mm, M/L+/-2 mm) and a Dremel with a sterilized 0.6 mm tip was used to remove an approximately 2 mm circular region around the implant site. The skull was cleaned, dried, and scored crosswise with a scalpel. While continuously perfusing the tissue with ACSF, the dura and cortex were removed using a 26 GA blunt-end needle connected to a vacuum line until the fibers of the corpus callosum became visible, at which point a 30 GA blunt-end needle was used to carefully remove the fibers of the callosum until striatal tissue became visible under a dissecting microscope. A 1 mm or 1.8 mm diameter GRIN lens was then lowered into the aspiration site (D/V –2.4 mm), held in place using a pipette tip and vacuum line attached to the stereotax. The lens was secured to the skull using dental cement (Metabond) and cyanoacrylate glue. Buprenorphine SR (1 mg·kg$^{-1}$) was administered following surgery, as well as meloxicam (2x/day for 48 hr, 5 mg·kg$^{-1}$), dexamethasone (1x/day, 0.2 mg·kg$^{-1}$ for 7 days), and Sulfatrim (for 7 days in the drinking water). Mice were allowed to recover for 2–4 weeks before mounting the miniature microscope.

## Miniscope mounting and imaging

Two to four weeks following surgery to implant the GRIN lens, the mouse was again anesthetized under the stereotax and a miniscope (UCLA V3 or V4) with a baseplate attached was lowered to the top focal plane of the GRIN lens. Once the optimal imaging plane was found, the baseplate was secured with dental cement (Metabond, Parkell).

## In vivo $Ca^{2+}$ imaging and analysis

All mice were habituated to the weight of the miniscope over a 60 min session in the open field. Following habituation, we placed mice in the open field for 60 min each day for 1–3 days with the miniscope attached. The number of cells and the number of sessions used per mouse for each analysis are contained in the Supplementary files (*Supplementary file 1*, *Supplementary file 2*, *Supplementary file 3*, *Supplementary file 4*). For analysis of calcium dynamics in dSPN cKO mice, we analyzed data from the first 20 min of day 1 of these sessions. In the majority of animals used in these experiments, we used v3 miniscopes which required higher LED intensities compared to v4 miniscopes. Therefore, we limited recording to 20 min to limit photobleaching. For experiments testing fenobam, JZL, and JNJ, 60 min of open field activity were recorded following IP injection. For these experiments, we used v4 scopes which allowed for the use of lower LED intensities. Fenobam (26 mg/kg) and JZL (10 mg/kg) were administered immediately prior to the test, following a 15 min habituation. Due to its pharmacokinetics, JNJ was administered 30 min prior to beginning the test. Fenobam was diluted in 40% beta-cyclodextrin, 10% DMSO, JZL, and JNJ were diluted in 10% DMSO and 90% corn oil.

We used open-source V3 and V4 miniscopes (https://github.com/Aharoni-Lab/Miniscope-v4; *Aharoni, 2023a*) that were connected to a coaxial cable which connected to a miniscope data acquisition board (DAQ) 3.3. The DAQ was connected to a computer via a USB 3.0 cable. Data was collected via the Miniscope QT Software (https://github.com/Aharoni-Lab/Miniscope-DAQ-QT-Software; *Aharoni, 2023b*) at 20 frames per second. Simultaneous video recordings of behavior were performed at approximately 30 fps, and data was aligned post hoc using the time stamps collected by the miniscope software for each frame. Miniscopes and DAQ boards were purchased from Open Ephys, LabMaker, or parts were machined by Shylo Steitler (UCLA) and assembled.

## Pre-processing and motion correction

Videos were preprocessed to remove sensor noise using a 2D FFT spatial frequency filter and low-pass Butterworth filter (https://github.com/Aharoni-Lab/Miniscope-v4/tree/master/Miniscope-v4-Denoising-Notebook) and then motion correction was performed using the normcorr algorithm in the CalmAn package (*Giovannucci et al., 2019*).

### Identification of neurons (CNMF-E)

Cells were automatically identified using the CNMF-E algorithm (*Zhou et al., 2018*). Recordings were grouped in batches of 1–5000 frames for processing and spatially downsampled 2x during the running of the CNMF-E algorithm. The maximum neuron diameter was determined based on manual observation of each of the motion-corrected videos in ImageJ and the minimum local correlation value (Cn) and signal-to-noise ratio (SNR) for a pixel to be used as a cell 'seed' was determined by visualizing the correlation image for a subset of recording frames of each video. The threshold for merging neurons was set at 0.65, neurons were merged if their distances were smaller than two pixels and they had highly correlated spatial shapes (correlation >0.8) and small temporal correlations (correlation <0.4). The output of the CNMF-E analysis performed on the video was then manually checked to ensure accuracy.

### Analysis of Ca$^{2+}$ activity

The cell Ca$^{2+}$ activity traces from CNMF-E were processed first by scaling to the 99-percentile value of each cell's activity trace during the recording session. The resulting normalized trace was then expressed as a Z-score by subtracting each sample from the mean and dividing by the standard deviation of the entire trace. Finally, this trace was downsampled to 5 Hz (using the maximum value within the downsampled regions) and this result was used for further analysis.

### Ca$^{2+}$ event rate and correlation to velocity

A threshold crossing algorithm was used to detect peaks that were ≥2.5 s.d. above the baseline signal value for 1 s and the time of each event was calculated as the midpoint between the maximum value of the event and the preceding minimum (*Parker et al., 2018*). We used EZ track (*Pennington et al., 2021*) to compute the frame-by-frame velocity of the animal in the open field arena. Using custom Python scripts, we applied a 1 s median filter to this velocity trace and downsampled to 5 Hz for alignment to the Ca$^{2+}$ activity data.

### Pairwise cell co-activity

The Jaccard index of pairwise coactivity was calculated following 1 s 'forward-smoothing' of the Ca$^{2+}$ activity traces as described previously (*Parker et al., 2018*). The mean value of the Jaccard index was calculated as a function of the Euclidean distance between cells in a neuron pair, and the mean values were binned into either 50 or 250 µm bins. These values were normalized to shuffled data sets where the Ca$^{2+}$ traces for each cell were shuffled in time, using 1000 independent shuffles of the binarized Ca$^{2+}$ event traces for each cell. This data was used for statistical comparison of coactivity between groups as a function of distance and behavioral state, i.e., moving or resting.

### Spatial coordination index

To measure spatial coordination, the extent to which co-activity is greater among more proximal neurons versus more distal cells, during each frame of the video, we calculated a previously used metric, the spatial coordination index (SCI) (*Parker et al., 2018*). For each frame of the down-sampled video, we shuffled the pairwise distances for each neuron pair (1000 independent shuffles for each frame of the video), but not each cell's Ca$^{2+}$ activity signal. From these 1000 different distributions of distances between cell pairs, we calculated the cumulative distribution of active cell distances for the actual dataset and each shuffled dataset. We then performed a pair of Kolmogorov–Smirnov tests to determine if the cumulative distribution of active cell distances in each frame was smaller (i.e. left-shifted or more 'clustered') or greater (right-shifted or less 'clustered') than that of the shuffled data. We took the logarithm (base ten) of the smaller p-value from either test and used this value as the spatial coordination index. If a frame did not have any active cell pairs, the index is 0. This analysis allows for visualization of how spatiotemporal dynamics change more specifically during movement onset.

### Detection of movement onset

To most precisely detect instances of movement onset, specifically forward ambulatory motion, for our SCI analysis, we used the frame-by-frame velocity calculated from the mouse's centroid location detected by ezTrack. We then filtered the velocity traces by determining if the animal's centroid

moved by more than 1 cm from its location in the current frame at any point over the next 1 s of the recording. If not, we set the velocity to 0 for that frame. We then smoothed the resulting trace using a 167 ms moving average. To find periods of movement onset that would allow us to more clearly measure whether there were consistent changes in spatial coordination, we identified where this velocity trace exceeded 0.5 cm/s for at least 2 s preceded by a period of rest for 4 s. We used the timestamps recorded by the miniscope software to find the matching period in our SCI signal. This allowed us to clearly measure how the degree of spatial coordination among dSPNs changes between rest and movement.

## Slice procedure and in vitro electrophysiological recording

Parasagittal slices containing the dorsal striatum were prepared using standard techniques. Mice were deeply anesthetized (xylazine 10 mg/kg and ketamine 100 mg/kg i.p.) before undergoing transcardial perfusion with an ice-cold sucrose artificial cerebrospinal fluid (ACSF) solution containing (in mM): 85 NaCl, 2.5 KCl, 1.25 $NaH_2PO_4$, 25 $NaHCO_3$, 25 glucose, 75 sucrose, 0.5 $CaCl_2$, and 4 $MgCl_2$, equilibrated with 95% $O_2$ and 5% $CO_2$. The brain was removed and mounted on the stage of vibratome (Leica Microsystems, Inc). 350 µm thick sections were made in ice-cold sucrose ACSF. Slices were transferred to a recovery chamber containing the sucrose slicing ACSF solution, which was gradually exchanged for a normal ACSF containing (in mM): 125 NaCl, 2.4 KCl, 1.2 $Na_2PO_4$, 25 $NaHCO_3$, 25 glucose, 1 $CaCl_2$, and 2 $MgCl_2$, while the slices were maintained at 30°C. Individual slices were transferred to a recording chamber and visualized under DIC. Micro-Manager open source software was used for collecting the CCD camera image (*Edelstein et al., 2014*). For extracellular recordings, slices were perfused with normal ACSF containing (in mM): 125 NaCl, 2.4 KCl, 1.2 $Na_2PO_4$, 25 $NaHCO_3$, 25 glucose, 2 $CaCl_2$, and 1 $MgCl_2$. Recording electrodes were manufactured from borosilicate glass pipettes and had resistances of 3–5 MΩ when filled with regular ACSF. For whole-cell voltage clamp experiments, recording electrodes were filled with internal solution containing (in mM) 95 CsF, 25 CsCl, 10 Cs-HEPES, 10 Cs-EGTA, 2 NaCl, 2 Mg-ATP, 10 QX-314, 5 TEA-Cl, 5 4-AP for recording of EPSCs, or 75 $CsCH_3SO_3$, 60 CsCl, 1 $MgCl_2$, 0.2 EGTA, 10 HEPES, 2 Mg-ATP, 0.3 GTP-$Na_2$, 10 $Na_2$-phosphocreatine, 10 TEA, 5 QX-314. SPNs were visually identified and were voltage clamped at –80 mV. Miniature events (mEPSCs) were recorded in the presence of the $Na^+$ channel blocker TTX (1 µM) and bicuculline (10 µM). Analysis of mEPSCs was performed using Eventer software (https://eventerneuro.netlify.app/).

## Histology

Following in vivo imaging experiments, mice were anesthetized using ketamine/xylazine and perfused with PBS containing 0.02% sodium nitrite and 4 mM $MgSO_4$ followed by 2% paraformaldehyde in 0.1 M sodium acetate buffer (pH 6.5) for 6 min, and 2% paraformaldehyde in 0.1 M sodium borate buffer (pH 8.5) for 12–18 min. The brains were removed from the skull, post-fixed overnight in 2% paraformaldehyde in 0.1 M sodium borate buffer (pH 8.5), then sectioned coronally at 50 µm on a Leica Vibratome VT1000s in PBS. The sections were collected and stored at 4°C in PBS containing 0.02% sodium azide. Free-floating sections were rinsed for 30 min in PBS followed by four 15 min washes in PBS containing 0.1% Triton X-100 (Sigma, T8787) and 0.1% bovine serum albumin (BSA; Sigma, A4503). The sections were then blocked with 3% normal donkey serum (NDS; Jackson ImmunoResearch, 017-000-121) for 1 hr and then incubated in primary antibody overnight (Mouse anti-GFP, Millipore, MAB3580). Sections were washed three times for 15 min and then incubated in secondary antibody for 1 hr and washed three times prior to mounting and imaging with a 20x objective on a Zeiss Imager M2 microscope.

## Statistics

For comparisons of single mean values per mouse, for instance, mean velocity per animal, we used non-parametric Mann-Whitney U test or the Wilcoxon Signed Rank test. Where the same measurement was made multiple times, we used a repeated measures ANOVA. For cases where multiple different but correlated, non-independent, comparisons were made within mice, for instance, in comparing mean event rates at different speeds or mean coactivity across neuron pairs at different distances, we used a linear mixed effects model (LMM) to account for the non-independence of measurements made from the same animal. The model was implemented using the statsmodels

Python package (https://www.statsmodels.org), the drug treatment or genotype was the independent variable, mean values measured per mouse are the dependent variables and individual mice are the group term. Where this model was used, we tested for the effect of genotype or drug treatment and reported the significance for that measurement. Results were considered significant when $p < 0.05$. In all figures, individual data points represent the mean of a group, and error bars represent the standard error of the mean. N numbers for each group are specified in the figure legend.

## Acknowledgements

The authors thank Dr. Craig Weiss and the Northwestern University behavioral phenotyping core facility for technical support.

## Additional information

### Funding

| Funder | Grant reference number | Author |
| --- | --- | --- |
| National Institute of Mental Health | R01MH099114 | Anis Contractor |

The funders had no role in study design, data collection and interpretation, or the decision to submit the work for publication.

### Author contributions

John J Marshall, Conceptualization, Data curation, Software, Formal analysis, Supervision, Funding acquisition, Investigation, Visualization, Methodology, Writing – original draft, Project administration; Jian Xu, Conceptualization, Resources, Investigation, Methodology, Writing – review and editing; Nai-Hsing Yeh, Toshihro Nomura, Investigation; Seongsik Yun, Formal analysis, Investigation, Methodology; John N Armstrong, Resources, Investigation; Jones Parker, Conceptualization, Resources, Software, Supervision, Methodology, Project administration, Writing – review and editing; Anis Contractor, Conceptualization, Resources, Formal analysis, Supervision, Funding acquisition, Visualization, Methodology, Writing – original draft, Project administration, Writing – review and editing

### Author ORCIDs

John J Marshall ⓘ https://orcid.org/0000-0002-6481-8413
Jones Parker ⓘ https://orcid.org/0000-0001-9302-1545
Anis Contractor ⓘ https://orcid.org/0000-0002-5131-2536

### Ethics

All experiments were performed in accordance with procedures and protocols approved by the Northwestern University IACUC.

Joint Public Review: https://doi.org/10.7554/eLife.98122.3.sa1
Author response https://doi.org/10.7554/eLife.98122.3.sa2

## Additional files

### Supplementary files

MDAR checklist

Supplementary file 1. Number of cells and recording sessions for analysis in *Figure 1*.

Supplementary file 2. Number of cells and recording sessions for analysis in *Figure 2*.

Supplementary file 3. Number of cells and recording sessions for analysis in *Figure 3*.

Supplementary file 4. Number of cells and recording sessions for analysis in *Figure 5*.

## Data availability

All data needed to evaluate the conclusions in the paper are present in the paper and/or the Supplementary Materials. The novel construct pAAV.Syn_hGH.DO_FAS.GCaMP6f.WPRE.bGHpA is available from the authors upon request. Source data for this study is available here: 10.5281/zenodo.17981142. Custom software code used for the analysis in this study is openly accessible here: (https://github.com/JMarshall-8413/Marshall_2025_SourceCode, copy archived at *Marshall, 2025*).

The following dataset was generated:

| Author(s) | Year | Dataset title | Dataset URL | Database and Identifier |
|---|---|---|---|---|
| Marshall J | 2025 | Source Data for "Synaptic mechanisms modulate the spatiotemporal dynamics of striatal direct pathway neurons and motor output" revision v2 | https://doi.org/10.5281/zenodo.17981142 | Zenodo, 10.5281/zenodo.17981142 |

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
