## [Editor Report · eLife Assessment]

Marshall et al describe the effects of altering metabotropic glutamate receptor 5 activity on activity of D1 receptor expressing spiny projection neurons in dorsolateral striatum focusing on two states - locomotion and rest. The authors examine effects of dSPN-specific constitutive mGlu5 deletion in several motor tests to arrive at this finding. Effects of inhibiting the degradation of the endocannabinoid 2-arachidonoyl glycerol are also examined. Overall, this is a **valuable** study that provides **solid** new information of relevance to movement disorders and possibly psychosis.

---

## [Referee Report · Joint Public Review]

Marshall et al describe the effects of altering metabotropic glutamate receptor 5 activity on activity of D1 receptor expressing spiny projection neurons in dorsolateral striatum focusing on two states - locomotion and rest. The authors examine effects of dSPN-specific constitutive mGlu5 deletion in several motor tests to arrive at this finding. Effects of inhibiting the degradation of the endocannabinoid 2-arachidonoyl glycerol are also examined. Overall, this is a valuable study that provides solid new information of relevance to movement disorders and possibly psychosis.

The combination of in vivo cellular calcium imaging, pharmacology, receptor knockout and movement analysis is effectively used. The main findings do not involve gross firing rates or numbers of active neurons, but rather are revealed by specialized measures involving Jaccard coefficient and an assessment of coactivity. The authors conclude that mGlu5 expressed in dSPNs contributes to movement through effects on clustered spatial coactivity of dSPNs. More specifically, reduced mGluR5 increases coactivity during rest (defined as low velocity periods) but not during locomotion periods. The authors observe a role for mGlu5 expression in dSPNs in modulating the frequency of mEPSCs, suggesting a role in presynaptic neurotransmitter release. Some data suggesting the story may be different in the other major SPN subpopulation (iSPNs) are also presented but these studies are relatively underdeveloped leaving some ambiguity as to how cell-selective the findings are. In addition, an occlusion experiment in which the pharmacological mGluR5 agents are delivered to the dSPN mGluR5 KO to clarify if other sites of action are involved beyond the proposed D1-expressing neurons is missing. Finally, the authors present a working model that sets the stage for future experimentation. Overall, this study provides an important and detailed assessment of mGluR5 contributions to striatal circuit function and behavior.

Remaining concerns include:

(1) To clarify that dSPNs are sole site of action, it is necessary to examine effects of the mGlu5 NAM in the dSPN mGlu5 cKO mice. If the effects of the two manipulations occluded one another this would certainly support the hypothesis that the drug effects are mediated by receptors expressed in dSPNs. A similar argument can be made for examining effects of the JNJ PAM in the cKO mice.

(2) There is a concern that the D1 Cre line used (Ey262), which may also target cortical neurons expands the interpretation of the study beyond the striatal populations. Further discussion of this point, particularly in the interpretation of the mGluR5 cKO experiments, would provide a better understanding of the contribution of the paper.

(3) The use of CsF-based whole-cell internal solutions has caused concern in some past studies due to possible interference with G-protein, phosphatase and channel function (https://www.sciencedirect.com/science/article/abs/pii/S1044743104000296, https://www.jneurosci.org/content/jneuro/6/10/2915.full.pdf). It is reassuring the DHPG-induced LTD was still observable with this solution. However, it might be worth examining this plasticity with a different internal to ensure that the magnitude of the agonist effect is not altered by this manipulation.

(4) Behavioral resolution of actions at low velocity that are termed "rest" are not explored in this study. Thus, a remaining ambiguity is whether the activities in rest include only periods of immobility or other low-velocity activities such as grooming or rearing.

---

## [Author Response]

The following is the authors’ response to the original reviews.

**Reviewer #1 (Public review):**
“Can the authors offer a hypothesis as to how decreased coactivity promotes increased movement velocity.”

In our revision we have added an additional metric measuring how spatial coactivity changes during movement onset, the spatial correlation index, which replicates a previous finding that co-activity among proximal neurons is statistically greater surrounding movement onset. We did not find, as outlined in the revision, that mGluR5 manipulations significantly altered this relationship. Our data therefore shows, consistent with that shown previously, that ensembles of dSPNs that are co-active during movement onset, in particular ambulatory movement, are more likely to contain neurons that are closer together and the neurons are highly active. In contrast, rest ensembles contain neurons that are less active but have more highly correlated activity, across all pairwise distances. Additionally, mGluR5 inhibition, genetic or pharmacological, promotes the activation of rest ensembles but does not affect the properties of movement ensembles. Previous studies (e.g. Klaus A. et al., 2017) have shown that neurons in rest ensembles are, in general, unlikely to also be members of movement ensembles, We therefore hypothesize that corticostriatal synapses onto SPNs of rest ensembles are more likely, during spontaneous behavior, to have reduced synaptic weight due to mGluR5 signaling, potentially due to eCB mediated inhibition of neurotransmitter release. Therefore, when we inhibit mGluR5 at these synapses, we increase synaptic weight and increase the probability of activation of this coordinated rest ensemble, which suppresses movement. If, on the other hand, the synapses that govern activation of neurons in movement ensembles have a higher weight, they may be unaffected by mGluR5 inhibition.

The use of the Jaccard similarity index in this study is not intuitive and not fully explained by the methods or the diagram in Figure 1.

We have added more detail to the paper to explain the methodology of the jaccard similarity measure. The advantage of this method is that is specifically captures cells that are jointly active, as opposed to jointly inactive and is therefore useful for capturing co-activity in our sparsely active Ca^2+^ imaging data.

The analysis of a possible 2-AG role in the mGlu5 mediated processes is incomplete.

We agree that, as an experiment to outline which endocannabinoids are involved in modulating synaptic strength through mGluR5, this experiment alone is not sufficient.

However, our main focus in this paper is how manipulations of mGluR5 affect the spatiotemporal dynamics of dSPNs and we chose not to focus on specific mechanisms of endocannabinoid signaling, though these would certainly be interesting to investigate further in vivo.

It would seem to be a simple experiment to examine effects of the mGlu5 NAM in the dSPN mGlu5 cKO mice. If effects of the two manipulations occluded one another this would certainly support the hypothesis that the drug effects are mediated by receptors expressed in dSPNs. A similar argument can be made for examining effects of the JNJ PAM in the cKO mice.

We agree that this experiment would be valuable and extend our findings presented in the paper, however, it has practically been outside the scope of the current work.

**Reviewer #2 (Public review):**
Pharmacological and genetic manipulations of mGluR5 do not differentially/preferentially modulate the activity of proximal vs distal dSPNs, therefore, it could also be interpreted that mGluR5 is blanketly boosting/suppressing all dSPN activity as opposed to differential proximal/distal spatial relationships.

As in the response to reviewer 1 above, we have added additional clarification to the text explaining that our manipulations do not differentially affect the co-activity of proximal vs distal dSPNs, this is also quantified throughout the text using the spatial coordination index. However, we disagree that “it could also be interpreted that mGluR5 is blanketly boosting/suppressing all dSPN activity” as we do not observe statistically significant changes in the event rate following either pharmacological or genetic manipulations of mGluR5. Rather, we consistently observe statistically significant changes in co-activity among neurons, the extent to which activity of active neurons during either rest or movement are correlated with each other. This is the central finding of our manuscript, inhibiting or potentiating mGluR5 signaling alters behavior, not by blanket suppression or enhancement of the activity as measured using the event rate, of dSPNs, but by affecting their ensemble dynamic properties. Co-activity during rest versus ambulatory movement is statistically greater in both proximal and distal cells and inhibiting mGluR5 increases this co-activity and decreases movement.

For these analyses of prox vs distal and all others, please include the detail of how many proximal vs distal cells were involved and per subject.

We have added a supplemental table that details the number of cells included per subject in all analyses

Ln. 151-152: Please provide data concerning how volumes of infectivity differ between injecting AAV vs. coating the lens? If these numbers are very different, this could impact the number of Jaccard pairings and bias results.

While viral injection may lead to a larger volume of expression, with this one photon imaging method only those cells within ~200 microns of the edge of the lens will be able to be resolved, therefore practically, if there is an additional volume of infected tissue outside of the field of view of the lens, it would not affect the results as these neurons will not be resolved by the endoscope camera. Accordingly, the average number of cells detected per session is very similar following each approach (mean # of cells per session with coating 90.93 ± 23.69 cells, with viral injection 90.03 ± 29.29 cells)

Is mGluR5 affecting dSPN activity in other measures beyond co-activity and rate? Does the amplitude of events change?

We have added supplemental data for figures 2, 3, and 5 demonstrating that manipulations of mGluR5 do not affect the amplitude or length of Ca^2+^ events included in the analysis.

What is the model of mGluR5 signaling in a resting state vs. movement? What other behaviors are occurring when the mouse is in a low velocity "resting state" (0-0.5 cm/s). If this includes other forms of movement (i.e. rearing, grooming) then the animal really isn't in a resting state. This is not mentioned in the open field behavior section of the methods and should be described (Ln. 486) in addition to greater explanation of what behavior measures were obtained from the video tracking software (only locomotion?)

It would be very interesting to determine if during “rest,” when the animals is not engaged in ambulatory behavior, it may be engaged in some fine motor behavior. However, the resolution of the cameras used to measure locomotor activity in this dataset does not allow us to do this.

There is large variability in co-activity in proximal dSPNs when animals are "resting" (2j). Could this be explained by different behavior states within your definition of "rest"?

We agree that if the animal is engaging in fine motor behavior that we cannot resolve with our behavior setup, this could produce some variability in coactivity. However, as shown previously (e.g. Klaus A. et al., 2017), ensembles active when the animal is not moving (our definition of “resting”), regardless of additional fine motor behaviors the animal may be engaged in when not moving, are substantially different that those ensembles that are active when the animal is moving. We therefore expect that this may limit, although potentially not eliminate, variability due to different behavioral states we may have grouped into our “resting” category. Unfortunately, as mentioned above, we are not able resolve variations in fine motor output in this behavioral data.

Have you performed IHC, ISH or another measure to validate D1 cell specific cKO?

The mGluR5^loxP/loxP^ mice used in this study were characterized previously by our lab (Xu et al., 2009), we used the same mice here with a different, but also published and characterized Cre-driver line, Drd1a-Cre Ey262 (Gerfen et al., 2013).

Why are the "Mean Norm Co-activity" values in 5e so high in this experiment relative to figures 2-4?

In experiments where we treated the same animal with vehicle and a drug (i.e., experiments in Figure 2 and 3), we normalized the values for each animal in the drug treatment group to the distal bin of that animal following vehicle treatment. This allowed us to more clearly resolve the changes within each animal due to drug treatment. As comparisons in the data in figure 5 d–f are between different animals (rather than different treatments of the same animal) we could not perform this normalization procedure.

**Reviewer #3 (Public review):**
Some D1 Cre lines have expression in the cortex. Which specific Cre line was used in this study?

We used, Drd1a-Cre Ey262. This is included in methods.

The text says JNJ treatment .... increased locomotor speed (Figure 3b) and increased the duration but not frequency of movement bouts (Figure 3c, d). However, the statistics of the figure legends say: however the change in mean velocity (3b) is not significant (p=0.060, U=3, Mann-Whitney U test), nor is the mean bout length during vehicle and JNJ (p=0.060, U=3, Mann-Whitney U test) (3d) Comparison of mean number of bouts of each animal during vehicle and JNJ (p=0.403, U=8, Mann-Whitney U test).

This has been corrected to indicate only the change in time spend at rest is statistically significant.

This effect was most pronounced during periods of rest (Figure 3i, j). The decrease was only in rest? Are the colors in Figure 3J inverted? Therefore, JNJ treatment had effects that were qualitatively the inverse to the effects of fenobam on locomotion and dSPN activity.

We have corrected the text to state that, overall, and during periods of rest but not movement, JNJ had effects that were qualitatively the opposite of fenobam.